# Nitrification of the lowermost stratosphere during the exceptionally cold Arctic winter 2015/16

Marleen Braun[1], Jens-Uwe Grooß[2], Wolfgang Woiwode[1], Sören Johansson[1], Michael Höpfner[1], Felix Friedl-Vallon[1], Hermann Oelhaf[1], Peter Preusse[2], Jörn Ungermann[2], Björn-Martin Sinnhuber[1], Helmut Ziereis[3], and Peter Braesicke[1]

[1]Institute of Meteorology and Climate Research, Karlsruhe Institute of Technology, Karlsruhe, Germany
[2]Institute of Energy- and Climate Research, Stratosphere (IEK-7), Forschungszentrum Jülich, Jülich, Germany
[3]Institute of Atmospheric Physics, German Aerospace Center, Oberpfaffenhofen, Germany

**Correspondence:** Marleen Braun (marleen.braun@kit.edu)

**Abstract.** The Arctic winter 2015/16 was characterized by exceptionally low stratospheric temperatures, favouring the formation of polar stratospheric clouds (PSCs) from mid-December until the end of February down to low stratospheric altitudes. Observations by GLORIA (Gimballed Limb Observer for Radiance Imaging of the Atmosphere) on HALO (High Altitude and LOng range research aircraft) during the PGS (POLSTRACC/GW-LCYLCE II/SALSA) campaign from December 2015 to March 2016 allow an investigation of the influence of denitrification on the lowermost stratosphere (LMS) with a high spatial resolution. Two-dimensional vertical cross-sections of nitric acid ($HNO_3$) along the flight track and tracer-tracer correlations derived from the GLORIA observations document detailed pictures of wide-spread nitrification of the Arctic LMS during the course of an entire winter. GLORIA observations show large-scale structures and local fine structures with enhanced absolute $HNO_3$ volume mixing ratios reaching up to 11 ppbv at altitudes of 13 km in January and nitrified filaments persisting until the middle of March. Narrow coherent structures tilted with altitude of enhanced $HNO_3$, observed in mid-January, are interpreted as regions recently nitrified by sublimating $HNO_3$-containing particles. Overall, extensive nitrification of the LMS between 5.0 ppbv and 7.0 ppbv at potential temperature levels between 350 and 380 K is estimated. The GLORIA observations are compared with CLaMS (Chemical Lagrangian Model of the Stratosphere) simulations. The fundamental structures observed by GLORIA are well reproduced, but differences in the fine structures are diagnosed. Further, CLaMS predominantly underestimates the spatial extent of $HNO_3$ maxima derived from the GLORIA observations as well as the overall nitrification of the LMS. Sensitivity simulations with CLaMS including (i) enhanced sedimentation rates in case of ice supersaturation (to resemble ice nucleation on NAT), (ii) a global temperature offset, (iii) modified growth rates (to resemble aspherical particles with larger surfaces) and (iv) temperature fluctuations (to resemble the impact of small-scale mountain waves) slightly improved the agreement with the GLORIA observations of individual flights. However, no parameter could be isolated which resulted in a general improvement for all flights. Still, the sensitivity simulations suggest that details of particle microphysics play a significant role for simulated LMS nitrification in January, while air subsidence, transport and mixing become increasingly important for the simulated $HNO_3$ distributions towards the end of the winter.

# 1 Introduction

The processes of denitrification and nitrification are well-known phenomena occurring in the polar winter stratosphere (Fahey et al., 1990). They involve the condensation, growth, sedimentation and sublimation of nitric acid ($HNO_3$)-containing polar stratospheric cloud (PSC) particles and result in an irreversible vertical redistribution of $HNO_3$. Denitrification is known to affect polar winter ozone loss (Fahey et al., 1990; Waibel, 1999). Denitrification at higher layers (i.e. around 16 to 22 km) attenuates fast deactivation of catalytically active chlorine species into the reservoir species chlorine nitrate ($ClONO_2$). However, chlorine deactivation can be fostered at lower layers enriched in $HNO_3$ (i.e. nitrified) by evaporating nitric acid trihydrate (NAT) particles (Fischer et al., 1997). Observational evidence for particles other than NAT involved in denitrification is sparse (e.g. Tabazadeh and Toon, 1996; Kim et al., 2006). Nitrification of the lowermost stratosphere is of particular interest since the chemical budget of reactive nitrogen ($NO_y$) and, thereby, its possible effects on ozone are modified in a region important for the radiative budget of the atmosphere (Riese et al., 2012).

While fundamental processes in connection with PSCs are well understood, there are still uncertainties concerning the formation of NAT particles and their characteristics that influence the processes of denitrification and nitrification. Chemistry-transport and global chemistry models including simplified microphysical properties of NAT are generally successful in simulating denitrification of the polar winter stratospheres (Carslaw, 2002; Grooß et al., 2005, 2014; Khosrawi et al., 2017; Zhu et al., 2017, and references therein). However, parametrizations resulting in agreement with observed size distributions of NAT particles, particularly extremely large NAT particles, and reproducing fine-structures of observed denitrification patterns remain an issue (e.g. Molleker et al., 2014; Woiwode et al., 2014).

Hemispheric differences in nitrification are observed due to different conditions in Antarctic and Arctic winter vortices. In the Antarctic, cold and stable vortices result in widespread PSC coverage and denitrification over wide vertical ranges. PSCs are observed less frequently in the Arctic, and the degree of denitrification varies from winter to winter (Santee et al., 1999; Pitts et al., 2018, and references therein). Increasing greenhouse gas emissions could lead to lower stratospheric temperatures (e.g. Rex et al., 2006) which likely cause stronger denitrification of the Arctic stratosphere.

Numerous studies addressed denitrification in Arctic winters, especially in cold winters, e.g. 1994/1995 (Waibel, 1999), 1999/2000 (Popp et al., 2001), 2004/2005 (Jin et al., 2006), 2009/2010 (Khosrawi et al., 2011; Woiwode et al., 2014), 2010/2011 (Sinnhuber et al., 2011) and 2015/2016 (Khosrawi et al., 2017). While most of these studies focused on the denitrification at altitudes higher than roughly 15 km, less attention was paid to the associated nitrification of lower layers. Dibb et al. (2006) reported nitrification at potential temperature levels above 340 K (around 12 km) at the end of January 2005. Further, Hübler et al. (1990) interpreted enhanced mixing ratios of up to 12 ppbv at altitudes between 10 and 12.5 km in the Arctic winter 1988/89 as resulting from nitrification. Tuck et al. (1997) found indications for nitrification in the Antarctic at levels above 400 K in the Antarctic winter 1994. Particularly, nitrification of the Arctic lowermost stratosphere (LMS) has hardly been investigated. This is due to the fact that cold winters with strong denitrification were rare events in the Arctic stratosphere in the past and that the observational capabilities to resolve nitrification of the LMS with sufficient coverage and vertical resolution are sparse. For example, limb-sounders, like MLS (Microwave Limb Sounder; Waters et al., 2006) or

MIPAS (Michelson Interferometer for Passive Atmospheric Sounding; Fischer et al., 2008) typically have vertical resolutions of around 3-5 km making it difficult to resolve fine-scale structures of $NO_y$ redistribution.

The process of vertical $HNO_3$ redistribution is very sensitive to temperature. NAT particle nucleation may begin as soon as temperatures fall below NAT equilibrium temperature $T_{NAT}$. However, clear evidence of the precise nucleation conditions of NAT particles is still lacking. NAT particles are nucleated heterogeneously with low number densities on foreign nuclei such as meteoritic dust particles (Hoyle et al., 2013). Below $T_{NAT}$, these particle grow and sediment downward and they evaporate as temperatures rise above $T_{NAT}$. A simulation of this process is challenging as it depends both on the nucleation parametrisation and on the precise reproduction of the temperatures around $T_{NAT}$. This is especially the case during the onset of this process. At a later time, to nucleate new NAT particles in denitrified air, lower temperatures are needed because of the already decreased $HNO_3$ mixing ratios. This results in a maximum potential denitrification for a given temperature. Since both the nucleation process and the mesoscale temperature modulations (e.g. by gravity waves) are not well known, it is especially difficult to simulate the small-scale structure of $HNO_3$ during the onset period.

Here, we present observations of nitrification of the LMS in the unusually cold Arctic winter 2015/2016 by the airborne limb-imaging Fourier transform infra red (FTIR) spectrometer GLORIA (Gimballed Limb Observer for Radiance Imaging of the Atmosphere, Friedl-Vallon et al. (2014); Riese et al. (2014)). In that winter, an extraordinarily cold and stable polar vortex (Matthias et al., 2016; Manney and Lawrence, 2016) promoted a long-lasting PSC phase from mid-December until the end of February with a large vertical extent (Pitts et al., 2018; Voigt et al., 2018) reaching down into the LMS.

Using the GLORIA observations and simulations by the Chemical Lagrangian Model of the Stratosphere (CLaMS; Grooß et al., 2014, references therein), we investigate following research questions:

- How are $HNO_3$ distributions structured in the LMS during the course of the cold Arctic winter 2015/16? How do $HNO_3$ distributions, which are affected by nitrification, compare with the stratospheric tracer ozone? How do observed small-scale spatial patters compare with a model (CLaMS)?

- Do tracer-tracer correlations constructed from GLORIA $HNO_3$ and $O_3$ indicate nitrification of the LMS? How does nitrification inferred from the GLORIA observations compare with that inferred from CLaMS? Can we identify a critical model parameter which results in a significant overall improvement of the agreement?

Thereby, we attempt to quantify the observed nitrification, which is particularly difficult because the LMS composition is influenced by air masses originating from the Arctic vortex, the extra-vortex stratosphere and the troposphere (Werner et al., 2010; Gettelman et al., 2011; Krause et al., 2018). $ClONO_2$ also contributed significantly to lowermost stratospheric total $NO_y$ during the Arctic winter 2015/16. This aspect is addressed within a separate study by Johansson et al. (2019). Here we focus only on gas-phase $HNO_3$, which is the direct product of nitrification by sublimating NAT particles.

We compare the GLORIA data with simulations by the CLaMS. To test how well different parametrizations within the same model reproduce the GLORIA observations, four sensitivity studies were performed. Those sensitivity simulations investigated the impact of (i) enhanced sedimentation rates in case of ice supersaturation, (ii) a global temperature offset, (iii) modified growth rates and (iv) temperature fluctuations.

## 2 Aircraft Campaign and Data

### 2.1 POLSTRACC/GW-LCYCLE II/SALSA

The GLORIA observations analysed in this study were obtained during the combined POLSTRACC (POLar STRAtosphere in a Changing Climate), GW-LCYCLE II (Gravity Wave Life Cycle Experiment) and SALSA (Seasonality of Air mass transport and origin in the Lowermost Stratosphere using the HALO Aircraft) campaigns (PGS). Starting from Oberpfaffenhofen, Germany or Kiruna, Sweden 18 research flights were carried out by the German research aircraft HALO (High Altitude and LOng range research aircraft) between December 2015 and March 2016. The flights probed an entire winter period in the LMS at high northern latitudes. For this study five research flights between December and March were used. The selection of the flight data was based on data availability and scientific requirements. Data availability was limited to flight sections where GLORIA was operated in the high spectral resolution "chemistry mode" (Friedl-Vallon et al., 2014) used in this study (see Sect. 2.2) and sufficiently cloud-free conditions allowing for the retrieval of $HNO_3$. From the scientific point of view, flights with long continuous "chemistry mode" measurements were chosen to show how patterns in the lowermost stratospheric $HNO_3$ distribution change during the winter. We furthermore focus on flights in January, where PSCs reached down to the LMS and where the most notable changes are found in the observed $HNO_3$ distributions. Since we use ozone as stratospheric tracer to quantify nitrification, flights in January are preferable since only little chemical ozone loss was diagnosed at this time of the winter when compared to February and March (see Johansson et al., 2019). Further GLORIA "chemistry mode" observations can be found in the supplementary information ofJohansson et al. (2018) and at the HALO Database (https://halo-db.pa.op.dlr.de/).

### 2.2 GLORIA

GLORIA is an airborne infrared limb imaging spectrometer (Friedl-Vallon et al., 2014). During PGS, GLORIA has been operated on board the HALO aircraft and pointed to the right hand side of the flight path. GLORIA combines a Michelson interferometer with an imaging HgCdTe detector which records 128 vertical and 48 horizontal interferograms simultaneously. All interferograms are transformed into spectra. The spectra from horizontal detector rows are averaged for noise reduction prior to the atmospheric parameter retrieval (Kleinert et al., 2014). In high spectral resolution mode, which is used in this study, the spectrometer covers the range from 780 to 1400 cm$^{-1}$ with a spectral sampling of 0.0625 cm$^{-1}$. For the retrieval, the radiative transfer code KOPRA (Karlsruhe Optimized and Precise Radiation transfer Algorithm; Stiller et al., 2002) and the inversion tool KOPRAFIT (Höpfner et al., 2001) were used. Estimated uncertainties of the GLORIA retrieval results are typically 1 - 2 K for temperature and 10 - 20% for trace gases. Typical vertical resolutions of the retrieved profiles are about 400 m at flight altitude and decrease to about 1000 m around the lowest tangent points. A detailed description and validation of the dataset used in this study is given by Johansson et al. (2018).

## 2.3  CLaMS

The Chemical Lagrangian Model of the Stratosphere (CLaMS) (McKenna, 2002a, b) is a chemistry transport model based on trajectory calculations for an ensemble of air parcels. CLaMS includes modules simulating Lagrangian trajectories, mixing, chemical processes and Lagrangian particle sedimentation. The CLaMS simulations used here were performed with a special setup for the POLSTRACC campaign with a horizontal resolution of about 100 km and a vertical resolution of about 500-900 m in the lower stratosphere above 10 km altitude decreasing to about 2 km below 9 km altitude. Further, this configuration includes a comprehensive stratospheric chemistry as described by Grooß et al. (2014). The simulations were performed for the entire winter and were based on meteorological wind and temperature data from the ECMWF ERA interim reanalysis (Dee et al., 2011) employing a horizontal resolution of 1x1 degrees and a timestep of 6 h. To simulate processes connected to NAT particles, particle parcels are implemented (Grooß et al., 2005, 2014). Particle size and number concentration are assigned to each particle parcel so that various particle parcels in one air parcel determine the particle size distribution. NAT and ice nucleation is temperature and saturation dependent and is parametrized by the scheme by Hoyle et al. (2013) and Engel et al. (2013), respectively. Particle growth and evaporation are calculated along particle trajectories based on Carslaw (2002) assuming the characteristics of spherical particles (Tritscher et al., 2019). Comparisons with PSC observations (Tritscher et al., 2019) show that the parametrisation of nucleation and sedimentation of NAT and ice particles in CLaMS is capable to reproduce the main features of PSC observations. Also, vortex averages of the vertical redistribution of $HNO_3$ and $H_2O$ have been reproduced (Tritscher et al., 2019).

## 3  Methods

### 3.1  GLORIA vertical cross sections of atmospheric parameters

The GLORIA retrieval results in vertical profiles of atmospheric parameters. These vertical profiles are combined to 2-dimensional quasi-vertical cross sections along the flight paths and show mesoscale atmospheric structures (Johansson et al., 2018). Since the observations are performed in limb-mode, the distance of the tangent points (i.e where the major information about atmospheric parameters stems from) gradually increases from the observer for the lower limb views. This is reflected by the tangent point distributions discussed in sections 4.1 to 4.3. The GLORIA data is filtered for cloud-affected observations, and data points with a vertical resolution worse than 2 km or above flight altitude are neglected for further analysis.

### 3.2  Simulated cross sections from CLaMS

For comparison with GLORIA, the CLaMS data were interpolated to the retrieval grid geolocations, characterized by altitude, latitude, longitude and time of the tangent points. The temporal interpolation with respect to atmospheric dynamics is performed by trajectory calculations. CLaMS output is typically saved daily at 12:00 UTC. Therefore forward trajectories are calculated for points between 00:00 UTC and 12:00 UTC until 12:00 UTC. The corresponding 12:00 UTC volume mixing ratio is then assumed as concentration of the original geolocation based on the assumption that chemical and physical changes in volume

mixing ratios during the time of the trajectory calculations are negligible for the chemical species considered here. For points between 12:00 UTC and 00:00 UTC backward trajectories are calculated analogously.

## 3.3 Identification of sub-vortex air

The altitude range of GLORIA observations in this study typically lies within the LMS, ranging from the tropopause to the 380 K isentrope (see Werner et al., 2010, and references therein). It has to be pointed out that robust identification of vortex air in the LMS is not possible due to dynamical disturbances, transport and in-mixing of air masses from different origins. In fact, the sub-vortex region in the LMS has a more filamentary character and is affected by interaction with air masses from the extra-vortex stratospheric overworld, the extra-tropical transition layer (ExTL), and the troposphere (Gettelman et al., 2011). Two filters have been applied to select data points associated with the sub-vortex region.

The first filter applied is the criterion by Nash et al. (1996) at the 370 K isentrope and based on the PV field obtained from the MERRA-2 reanalysis (Gelaro et al., 2017). Grid points with latitude-longitude pairs outside the polar vortex at the potential temperature ($\theta$) level of 370 K are classified as non-vortex points. Secondly, data is filtered by scaled potential vorticity (sPV) calculated from MERRA-2 reanalysis data with a threshold of $1.2 \times 10^{-4} s^{-1}$. sPV is calculated by dividing the potential vorticity (PV) by $\partial\theta/\partial p$ to obtain similar PV ranges for all isentropic levels that are investigated (Manney et al., 1994; Dunkerton and Delisi, 1986). Therefore this filter takes the altitude information of the grid points into account. Data points in the tracer correlations (see below) are attributed to sub-vortex air, if both criteria are met.

## 3.4 Quantification of nitrification based on tracer-tracer correlations using relative normalized frequency distributions

To quantify nitrification in the LMS tracer-tracer correlations of $HNO_3$ and $O_3$ associated with sub-vortex air are analysed. Here, we use ozone as an approximation of a passive reference tracer, since ozone is well accessible with GLORIA and shows a sufficient vertical gradient in the LMS region. The choice of ozone as a passive tracer is based on the assumption that ozone depletion is small in January as the air is hardly exposed to sunlight. The model study by Khosrawi et al. (2017) supports this assumption. Two aspects can affect the correlation: 1) Mixing with extra-vortex air masses not affected by nitrification would lead to an underestimation of $HNO_3$ introduced into the LMS by nitrification and 2) Potential ozone depletion would shift higher $HNO_3$ mixing ratios to lower ozone values, thus enhancing estimated nitrification.

As correlation scatter plots of measured data for several flights are difficult to assess due to the large number of individual points, estimates of relative normalized frequency distributions (RNFD) as described by Eckstein et al. (2018) are used in this study. This method calculates a scaled two dimensional histogram on a volume mixing ratio grid. In this study a grid of 0.070 ppmv $O_3$ × 0.35 ppbv $HNO_3$ is chosen, which is motivated by the total estimated error of the trace gases (Johansson et al., 2018). GLORIA data points with a calculated relative error larger than 20% are neglected in this study. Besides a clearer presentation of the data points, RNFDs filter out single data points with very high $HNO_3$ volume mixing ratios. Therefore, in the context of a challenging vortex identification this method offers an additional filter, as single data points that are differing significantly and are erroneously identified as vortex air are filtered out. Here it has to be pointed out that also local non-

erroneous points with very high HNO$_3$ values within the vortex are filtered out applying this method. However, in this study we aim to quantify the overall nitrification of the LMS, while local nitrification is highly inhomogeneous and can reach

10   significantly higher values. The RNFD contour line used for quantification in this study includes points within 2% of the histograms maximum density. An example of a RNFD for the HNO$_3$-O$_3$-correlation during flight 8 is given in Figure 1. We also show alternative isolines including 1 % and 4 % of the maximum density of the histogram to visualize weaker and stronger thresholds for statistical outliers. In all cases, the isolines show a similar pattern in general. However, stronger threshold values limit the vertical range of the analysis and filter out valuable significant data points.

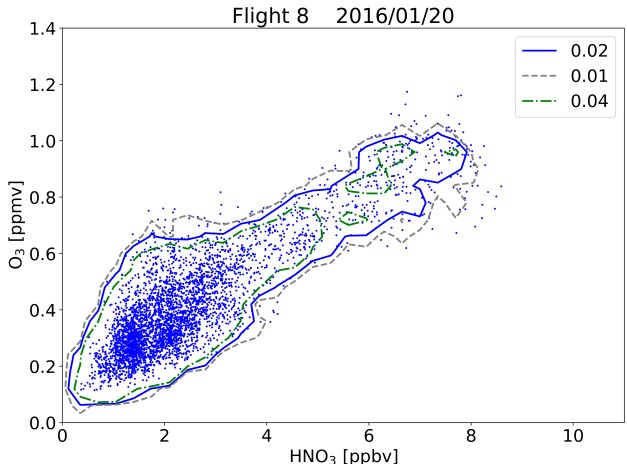

**Figure 1.** Correlation of HNO$_3$ with O$_3$ measured by GLORIA during flight 8 on 20 January 2016. The correlation is shown as single points and as RNFD (solid, dashed, and dash-dotted lines; see legend).

15  **4   GLORIA observations and CLaMS simulations of selected flights from January to March 2016**

To investigate how the observed HNO$_3$ distribution is affected by nitrification, three research flights have been selected. The first flight was carried out on 20 January 2016 during the coldest phase of the winter (Manney and Lawrence, 2016), with PSCs ranging down to flight level. The second flight took place on 31 January 2016 after a strong PSC phase. The last flight was carried out on 18 March 2016 at a late state of the winter - about two weeks after the final warming (5-6 March; Manney and Lawrence, 2016).

The observed patterns in the HNO$_3$ distributions are compared with the observed patterns in the ozone distribution. Since ozone and HNO$_3$ are effected by the same dynamical processes, the different patterns in the observed distributions are likely caused by processes that effect only one species (i.e. nitrification due to sublimation of NO$_y$-containing particles sedimented from higher altitudes). Therefore, the local HNO$_3$ enhancements seen in comparing adjacent air masses at a given height level and the deviations of their pattern from the pattern seen in the ozone distribution are interpreted qualitatively as a result of

nitrification.

## 4.1 Flight 8 on 20 January 2016

Applying the Nash criterion on 20 January 2016, a relatively coherent vortex region is found, with all GLORIA tangent points located inside the vortex region at $\theta = 370K$ (Fig. 2a). Since clouds complicate a robust trace gas retrieval, a number of GLORIA observations were removed by cloud-filtering. As a consequence, only limited GLORIA nitric acid data are available in flight sections with sufficiently transparent conditions (Fig. 2b). Further, particulate $NO_y$ (i.e. the difference between measured total $NO_y$ and gas-phase $NO_y$) was simultaneously measured in-situ by using a chemiluminescence-detector in combination with a converter for $NO_y$ species (Stratmann et al., 2016). Similar observations have also been made during other aircraft campaigns in the Arctic (Northway et al., 2002).

The measurements shown in Fig. 2c are based on the sub-isokinetic sampling of particles with a forward looking inlet (e.g. Fahey et al., 1989; Ziereis et al., 2004). Particles larger than a few tenths of a micrometer are sampled with enhanced efficiency and are detected as gas-phase equivalent $NO_y^*$. The efficiency factor depends among others on the ratio between aircraft and sampling velocity, pressure, temperature, and particle size (e.g. Fahey et al., 1989; Feigl et al., 1999). Maximum enhancement factor may be achieved for particle sizes larger than about 10 μm and is on the order of several tens, depending on the actual combination of the above mentioned parameters. Here, only equivalent $NO_y^*$ that was not corrected for enhancement is shown as a qualitative proxy for particulate $HNO_3$. As absolute values can not be obtained, we only use the data as a proxy for condensed $HNO_3$-containing particles present at flight altitude. The in situ data clearly confirm the presence of $HNO_3$-containing PSC particles at flight altitude and in the vicinity of the local $HNO_3$ maxima detected by GLORIA at waypoint A and after waypoint B.

The vertical cross-sections of $O_3$ and $HNO_3$ volume mixing ratios along the HALO flight track derived from GLORIA are depicted in Fig. 2b, d. The ozone distribution shows increasing volume mixing ratios with altitude reaching 1.1 ppmv at 13 km. The observed $O_3$ volume mixing ratios vary only moderately at fixed altitude levels during the whole flight in agreement with the location of the measurements within the vortex and the homogeneity inside the vortex. Compared to the ozone distribution, the $HNO_3$ volume mixing ratios are varying significantly at fixed altitudes. The $HNO_3$ distribution shows high $HNO_3$ volume mixing ratios particularly in the flight segment between the waypoints B and C, reaching up to 8 ppbv at a flight altitude of 13 km compared to 3 ppbv observed in adjacent air around waypoint C. Differences of this maximum structure from the corresponding $O_3$ distribution are interpreted qualitatively as nitrified air. Further, local maxima are forming coherent structures tilted with altitude and are observed down to 11 km in that flight segment. In addition, small scale fine structures with enhanced $HNO_3$ volume mixing ratios appear between 15:30 and 17:00 UTC and reach down to 10 km. The pattern of continuous and slightly tilted vertical bands differ significantly from the ozone distribution and show enhanced values compared to adjacent air masses at a given height level, thus suggesting their formation by redistribution of $HNO_3$.

To test this hypothesis, we show normalized GLORIA $HNO_3$ and $O_3$ data along selected isentropes in Fig. 2f. Normalisation factors are chosen in a way such that the mixing ratios of both gases are close to 1 in air masses which are not affected by nitrification. In unaffected air masses, the normalized mixing ratios of these gases are expected to show the same pattern. In nitrified air masses, locally enhanced $HNO_3$ and different modulations are expected relative to $O_3$. In fact, such local maxima in

the HNO$_3$ mixing ratios can be identified at 340 K around 15:45, 16:25, 18:20 and 18:40 UTC and, more pronounced, at 350 K around 18:35 and after 19:10 UTC. The maxima clearly coincide with local maxima seen in the HNO$_3$ cross sections. Thus, the simultaneous presence of confined local gas phase HNO$_3$ maxima in the GLORIA data and HNO$_3$-containing particles detected in situ in regions close to the equilibrium temperature of NAT and well above the equilibrium temperature of ice (see GLORIA temperature data shown in Johansson et al., 2018) suggests that an ongoing nitrification process was probed.

The vertical cross-section of HNO$_3$ volume mixing ratios modelled by CLaMS is shown in Fig. 2e. HNO$_3$ volume mixing ratios reach maximum values of 8 ppbv at flight altitude in the flight segment between B and C. While maximum HNO$_3$ volume mixing ratios in this flight are well represented by CLaMS, slight differences in the location of the maximum occur. Overall HNO$_3$ mixing ratios are clearly underestimated by CLaMS, and CLaMS mostly misses the vertical fine structure.

## 4.2 Flight 12 on 31 January 2016

At the end of January 2016, applying the Nash criterion at $\theta = 370K$, a more disturbed lower vortex region is observed. As shown in Fig. 3a, a large region between Greenland, central Europe and northern Siberia fulfilled the vortex criterion. However, filaments of lower PV are found from Greenland to southern Scandinavia and around the eastern rim of Scandinavia. Flight 12 was carried out starting and ending in Kiruna on 31 January and intersected several times with filaments outside the vortex.

Measured cross-sections of O$_3$ and HNO$_3$ volume mixing ratios are depicted in Fig. 3b and 3c. Except for the flight segments between 8:40 UTC and 9:20 UTC as well as 11:30 UTC and 11:50 UTC, that are associated with vortex edge or non vortex air, ozone values increasing with height are observed. Compared to the ozone values only varying slightly along an isentrope, the HNO$_3$ volume mixing ratios show larger variations at levels of constant potential temperature and suggest local enhancements by nitrification. Again, the analysis of normalized HNO$_3$ and O$_3$ along the selected isentropes clearly shows enhanced and more variable HNO$_3$ mixing ratios relative to O$_3$ inside air masses attributed to the sub-vortex region (Fig. 3e). During this flight, high local maximum values well above 10 ppbv are found between 13:30 UTC and 14:30 UTC in the GLORIA observations.

The HNO$_3$ distribution modelled by CLaMS is shown in Fig. 3d. When compared to GLORIA, locally more confined and weaker HNO$_3$ maxima are modelled after 12:15 UTC reaching down to altitudes of 12 km. Maximum HNO$_3$ volume mixing ratios are found at flight altitude showing narrow peaks up to 10 ppbv at 12:15 UTC and 8 ppbv at 12:50 UTC, at waypoint C and D. Overall, CLaMS shows a higher spatial variability and underestimates the maximum values during large parts of the flight.

## 4.3 Flight 21 on 18 March 2016

For the flight on 18 March 2016 the PV distribution shows a patchy pattern of regions inside the remains of the vortex according to Nash et al. (1996) around Scandinavia, with the GLORIA observations being located partly inside and outside these regions (Fig. 4a).

The measured O$_3$ distribution (Fig. 4b) shows increasing values with altitude and reaches values of 1.2 ppmv at flight level. Ozone values along the isentropes vary only slightly. The measured HNO$_3$ distribution (Fig. 4c) shows a slightly higher variability along the isentropes with local maxima for altitudes higher than 9 km reaching maximum values of up to 6 ppbv at

flight altitude embedded in background values of 2 to 3 ppbv. Filamentation and mixing following the earlier vortex break-up (Manney and Lawrence, 2016) resulted in less spatial variability when compared to the previous flights. Since flight 21 was carried out after the vortex break-up and the correlation of $HNO_3$ and $O_3$ was altered by in-mixing of extra-vortex air, this flight is included in the model comparisons Sect. 6, but not in the quantification of nitrification of the LMS in Sect. 5. However, well-defined local maxima qualitatively attributed to result from nitrification by the comparison with the ozone distribution still persisted in this late stage of the winter. The analysis of normalized $HNO_3$ and $O_3$ along isentropes shows enhanced and slightly more variable $HNO_3$ mixing ratios relative to $O_3$ in the sub-vortex region and its vicinity, thus supporting that these patterns are remnants of nitrification (Fig. 4e).

CLaMS (Fig.4d) shows $HNO_3$ volume mixing ratios for altitudes higher than 9 km corresponding well with GLORIA observations. Maximum values of locally 6 ppbv are modelled around 12:30 UTC at 13 km. Again, CLaMS slightly underestimates overall $HNO_3$ mixing ratios when compared to GLORIA.

## 5  Quantification of nitrification of the LMS from December 2015 to January 2016

To quantify nitrification in the LMS from December 2015 to January 2016 we applied the method described in section 3.4 using selected flights in this period. Fig. 5a depicts the distributions for flights 5, 6, 8 and 12 derived from GLORIA observations. Flight 5 (only limited GLORIA data available, see Appendix A) was carried out on 21 December 2015, at the beginning of the winter, with no significant hints to nitrification. Therefore this flight was chosen as early winter reference. Due to a limited number of points associated with vortex air and since sub-vortex and non-vortex data points show a compact correlation for flight 5, non-vortex points are also included here to extent the available data. For all other flights, only data points associated with vortex air are used. Flight 6 (see Appendix A) covered a broad range of latitudes in the sub-vortex region and below PSCs (Pitts et al., 2018).

$HNO_3$ volume mixing ratios for flight 5 range up to 3.2 ppbv with an approximately linear relationship to the observed ozone values. In case of flight 6 enhanced $HNO_3$ volume mixing ratios compared to flight 5 are observed for all ozone values. Flight 8 shows a similar enhancement throughout the whole range of ozone mixing ratios observed. While the enhancement is similar to flight 6, minimum $HNO_3$ values for ozone values higher than 0.7 ppmv are higher than for the flights before. For flight 12, the $HNO_3$ volume mixing ratios reach higher values than for the previous flights. Altogether, comparing maximum values with the early winter reference, an ongoing nitrification is observed between December 2015 and January 2016 reaching up to 7 ppbv at ozone values of 1 ppmv and 5 ppbv at ozone values of 0.6 ppmv.

Johansson et al. (2019) estimated an ozone depletion by 0.15 ppmv at 380 K for ozone values around 1.15 ppmv by the end of January 2016. Assuming this potential ozone depletion of 15% (dashed profile in Figure 5a) in the LMS during the given time frame, the estimated nitrification would reduce to 6 ppbv at 1 ppmv $O_3$ and 3.5 ppbv at ozone values of 0.6 ppmv. This is a lower limit estimation, especially considering the contrary effect by mixing of non-vortex air masses.

Correlation-based approaches are also suitable for model comparisons. The exact reproduction of complex fine structures by models cannot be expected because of uncertainties in measurements and the model. Differences in the meteorological fields

used for modelling, especially the temperature, can result in differing local structures. Since the investigated flights probed a wide range of the subvortex region, the obtained correlations can be regarded as representative for the (Arctic) sub vortex and allow for a comparison between model and measurement. We point out that observed differences in RNFDs can be caused by an inaccurate representation of processes influencing both, $HNO_3$ as well as $O_3$ volume mixing ratios.

The distributions simulated by CLaMS are shown in Fig. 5b. For flight 5 the $HNO_3$ volume mixing ratios range up to 2.5 ppbv in an approximately linear relationship. Flight 6 shows enhanced $HNO_3$ volume mixing ratios that are significantly lower than for the GLORIA observations. CLaMS models a further enhancement for flight 8 with a small patch reaching the maximum values observed by GLORIA. Similar to the GLORIA measurements, flight 12 displays the highest $HNO_3$ volume mixing ratios of all flights. However, the maximum values observed by CLaMS are 2 ppbv lower. Beneath 0.3 ppmv $O_3$ hardly

any enhancement is observed over the duration of the flights. Overall, CLaMS is able to reproduce the general enhancement of $HNO_3$ during the winter leading to a nitrification of up to 4 ppbv for ozone values of 0.8 to 1 ppmv, which is by 3 ppbv $HNO_3$ lower than the GLORIA observations.

## 6    Comparison of GLORIA results with CLaMS sensitivity simulations

Four sensitivity simulations have been performed to investigate processes and aspects that have not been represented in the
model so far. These sensitivity simulations were performed based on assumptions concerning particle formation and shape. Besides the formation of NAT on ice particles, ice can possibly accumulate on NAT particles (Voigt et al., 2018) resulting in larger particles with an enhanced settling velocity. Therefore in the 'ice settling' simulation the computed ice settling velocity (computed as described by Tritscher et al., 2019) was increased by a factor of 1.5 at all locations where the saturation ratio of ice, $S_{Ice}$, is larger than 1.2. Since NAT formation is temperature dependent a sensitivity simulation is performed
with a global temperature offset of 1 K. Particle growth in CLaMS is based on the assumption of growth rates of spherical particles. However, Woiwode et al. (2016) found indications for highly aspherical particles with an enhanced surface compared to spherical particles of the same volume. Since the $HNO_3$ uptake depends on the surface, a faster particle growth would occur. A 1.5 times enhanced particle growth was implemented in the 'aspherical particle' simulation. Changes in settling velocities due to different shapes have not been taken into account here. Several studies suggest a connection between orographically induced gravity waves and NAT formation (Davies et al., 2005; Carslaw et al., 1998; Höpfner et al., 2006). However small scale temperature fluctuations are not resolved by ERA interim temperatures. Therefore, artificial fluctuations according to (Tritscher et al., 2019) have been added in the 'temperature fluctuations' simulation.

5    The comparison is based on the RNFDs depicted for the individual flights in Fig. 6. The model cross-sections compared to measurements of flights 6, 8, 12 and 21 can be found in the Appendix (Fig. B1 B2, B3, B4).

The 'ice settling' simulation (yellow) delivers nearly identical results as the reference simulation for all flights. For the 'T-1K' simulation (dark blue), enhanced $HNO_3$ volume mixing ratios are observed down to lower $O_3$ volume mixing ratios compared to the reference simulations for flight 6 and 8, but are still not reaching down to the ozone values noticed by GLORIA. The high
10    $HNO_3$ volume mixing ratios measured by GLORIA are still underestimated here. While there are only slight changes compared

to the reference simulation for flight 12, the 'T-1K' simulation is clearly deteriorating for flight 21. Here, lower ozone volume mixing ratios are observed. For this flight none of the CLaMS simulations is able to reproduce the high ozone volume mixing ratios observed by GLORIA, which is possibly caused by weaker subsidence in the model. Further, lower absolute $HNO_3$ values might occur due to stronger mixing in CLaMS. The 'aspherical particle' case results in enhanced values observed down to lower altitudes than for the reference simulation for the flights 6, 8 and 12. Further, it shows higher maximum $HNO_3$ values than for the reference for flights 6 and 8. However, the absolute values are still underestimated compared to the GLORIA observations. For the flights 12 and 21 indications for points with lower $HNO_3$ values compared to the reference are found. The RNFD structure of 'temperature fluctuation' simulation for flight 6 and 21 are nearly identical to the reference simulation. For flight 8, lower $HNO_3$ volume mixing ratios are observed. In contrast to that, the 'temperature fluctuation' simulation for flight 12 shows best agreement with the observations. However, even though the lower branch is consistent with GLORIA, an upper branch with values lower than the reference simulation and far lower than GLORIA exists.

## 7    Discussion and Conclusion

Nitrification of the LMS in the Arctic winter 2015/16 was analysed based on GLORIA measurements during the PGS campaign. Vertical cross sections of $HNO_3$ volume mixing ratios for several flights throughout the winter show complex fine scale structures and enhanced values at altitudes down to 9 km. Flight 8 on 20 January 2016 was carried out under cold conditions with PSC observations at flight altitude. For this flight, coherent structures tilted with altitude of locally enhanced $HNO_3$ volume mixing ratios are observed that most likely indicate defined regions where settled $HNO_3$-containing particles recently sublimated. This is supported by simultaneous in situ observations of $HNO_3$-containing particles. The net effect of proceeding nitrification and dynamical processes in the LMS is observed for flight 12 at the end of January with a pronounced pattern of enhanced $HNO_3$ volume mixing ratios well exceeding 10 ppbv. Nitrified filaments with $HNO_3$ volume mixing ratios up to 6 ppbv persist until flight 21 in March 2016. While cross-sections provide a qualitative insight on local nitrification patterns for selected flights, the extent of overall nitrification has been quantified based on $HNO_3$-$O_3$-correlations. Nitrification reached an extent of up to 7 ppbv at ozone values of 1 ppmv ($\theta \approx 370$ K) and up to 5 ppbv at ozone values of 0.6 ppmv ($\theta \approx 350$ K). A conservative correction, assuming a 15 % ozone loss on the correlations would reduce these numbers to 6 ppbv and 3.5 ppbv, respectively.

The comparison of GLORIA observations with the chemistry transport model CLaMS confirm the model's ability to reproduce nitrification of the LMS. Large-scale structures are reproduced by the model that also resolves complex fine structures, although differing from measured patterns. CLaMS predominantly underestimates the enhanced values observed by GLORIA. Enhanced values are found less frequently in the simulation and are limited to narrow regions. Further, modelled $HNO_3$ enhancements reach less far down on 12 and 20 January 2016 when compared with GLORIA. The CLaMS simulations result in a weaker nitrification of up to 4 ppbv for the period of December to January for ozone mixing ratios between 0.8 ppmv to 1 ppmv, which is by ~3 ppbv lower than observed by GLORIA. For flight 21 in March, CLaMS underestimates the observed ozone volume mixing ratios, potentially indicating insufficient subsidence and stronger mixing in the model (Johansson et al.,

2019). Sensitivity studies with CLaMS considering i) ice formation on NAT particles, ii) a 1 K global temperature offset, iii) growth rates of aspherical particles or iv) temperature fluctuations were performed. While the 'ice formation' simulation shows only slight differences, the other cases show noticeable differences during individual flights. The 'temperature fluctuation' simulation provides improved agreement for the flight on 31 January 2016, but also worsens the results for the flight on 20 January 2016. The 'T-1K' simulation improves the results for the flights 6 and 8 in January, but deteriorates the results for the flight in the late winter on 18 March 2016. This shows the sensitivity of the simulation results on temperature. Potentially, a higher resolution in time and space would result in more realistic temperature fluctuations and could improve the simulations. The 'aspherical particle' case shows slightly more pronounced improvements for the flights in mid-January. Even though the sensitivity simulations partially improve the results, distinct differences between model and measurements remain. Therefore, we conclude that a more comprehensive change in the model representations is required. However, the sensitivity simulations suggest that particle microphysics play a significant role for LMS nitrification in January. Increasing discrepancies from the observations towards the end of the winter are attributed to simulated air subsidence, transport and mixing processes.

Several studies investigated nitrification in previous cold winters, although mainly with a focus on higher altitudes. Hübler et al. (1990) interpreted enhanced $NO_y$ values of up to 12 ppbv at altitudes between 10 and 12.5 km in the Arctic winter 1988/89 as a result of nitrification. For the Arctic winter 2002/03 Grooß et al. (2005) modelled a nitrification of less than 1 ppbv for potential temperatures lower than 360 K. For the Arctic winter 2004/05, Dibb et al. (2006) observe nitrification of up to 3 ppbv for potential temperatures between 360 and 340 K. Jin et al. (2006) report an average nitrification of less than 2 ppbv for potential temperatures lower than 370 K for the same winter. Further, during the Arctic winter 2009/10 (Grooß et al., 2014) modelled a nitrification of less than 1 ppbv for potential temperatures lower than 360 K, while (Woiwode et al., 2014) found no significant indications for nitrification below 370 K. Since Arctic winters might show a tendency towards colder stratospheric temperatures (Rex et al., 2006), disturbances of the LMS $NO_y$ budget by nitrification are likely becoming more frequent. During the Arctic winter 2015/16 exceptionally low stratospheric temperatures occurred and the vortex was sufficiently stable to allow formation of PSCs down to lowest stratospheric altitudes. Those conditions were the prerequisites for the strong nitrification observed and presented here. The measurements obtained by GLORIA during the POLSTRACC campaign document in detail a strong impact of nitrification on the LMS during an entire Arctic winter.

*Data availability.* The discussed GLORIA data set is available at the HALO database at https://halo-db.pa.op.dlr.de/ and at the KITopen repository (https://doi.org/10.5445/IR/1000086506). NASA MERRA-2 reanalysis data is available at https://disc.gsfc.nasa.gov/.

**Appendix A**

Figure A1a shows the flight track and GLORIA tangent of points of flight 5 on 21 December 2015. The flight accessed air masses associated with the sub-vortex and its vicinity in the region around Scandinavia. Figures A1b and A1c show the associated vertical cross-sections of $O_3$ and $HNO_3$ derived from the GLORIA observations.

Figure A2a shows the flight track and GLORIA tangent of points of flight 6 on 12 January 2016. The flight crossed the polar front jet stream above Italy and accessed sub-vortex air masses between northern Italy and Scandinavia. The associated vertical cross sections of $O_3$ and $HNO_3$ derived from the GLORIA observations are shown in Figure A2b and A2c.

15  **Appendix B**

*Author contributions.* MB conducted the analysis and interpretation of GLORIA level-2 data and model simulations and prepared the manuscript with contributions from all co-authors. JUG performed the CLaMS model simulations. SJ, JU, WW performed the level-1 and -2 analysis of GLORIA data. MH contributed to the GLORIA data analysis and interpretation. FFV, PP coordinated the GLORIA operations during the PGS campaign. HO, BMS coordinated the PGS field campaign. HZ provided the particle $HNO_3$ data. PB contributed to the interpretation and the manuscript preparation.

*Competing interests.* The authors declare that they have no conflict of interest.

*Acknowledgements.* We thank the PGS coordination team and the DLR-FX for successfully conducting the field campaign. The results are based on the efforts of all members of the GLORIA team, including the technology institutes ZEA-1 and ZEA-2 at Forschungszentrum Jülich. We thank NASA for providing their MERRA-2 meteorological reanalysis data set. We acknowledge the computing time for the CLaMS 5 simulations granted on the supercomputer JURECA at Jülich Supercomputing Centre (JSC) under the VSR project ID JICG11. This work was supported by the German Research Foundation (Deutsche Forschungsgemeinschaft, DFG Priority Program SPP 1294).
Research by W. Woiwode is supported by the DFG grant WO 2160/1-1. S. Johansson has received funding from the European Community's Seventh Framework Programme (FP7/2007-2013) under grant agreement 603557. Further support was received by the German research initiative ROMIC (Role of the Middle Atmosphere in Climate) and by the German Ministry of Research and Education (BMBF) project "Investigation of the life cycle of gravity waves" (GW-LCYCLE, subproject 2, 01LG1206B). We acknowledge support by the Deutsche Forschungsgemeinschaft and the Open Access Publishing Fund of the Karlsruhe Institute of Technology.

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

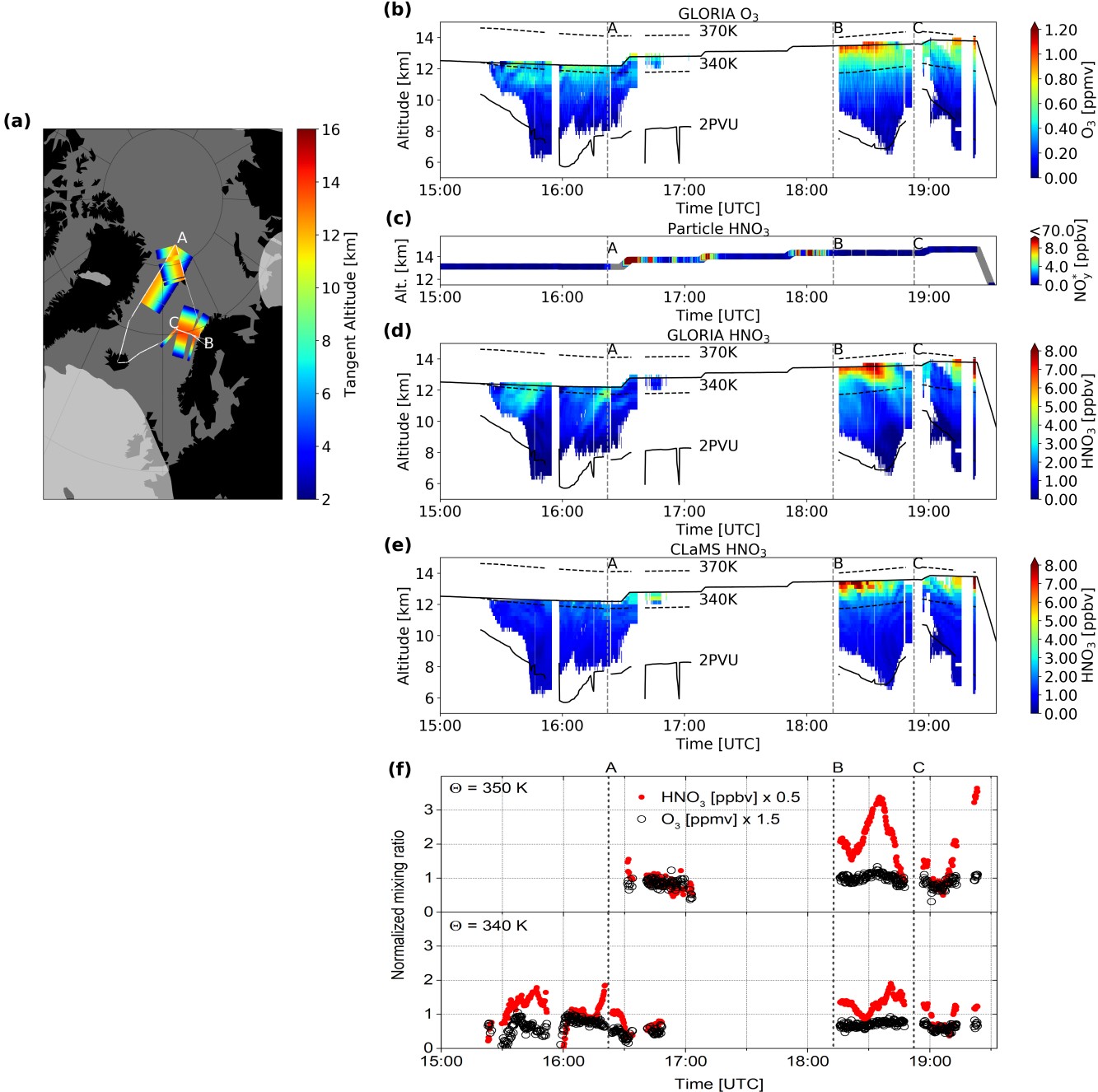

**Figure 2.** (a) Flight path and vortex filtering according to the Nash criterion at 370 K for flight 8 on 20 January 2016. White line: flight track with characteristic waypoints (A, B, C); dark grey shading: areas that are associated with the polar vortex. (b) Cross-section of $O_3$ distribution derived from GLORIA. Flight altitude (bold black line), characteristic waypoints (A, B, C), 340 K and 370 K potential temperature levels (MERRA-2, dashed black lines) and 2 PVU level (MERRA-2, black line). (c) In situ measurements of gas-phase equivalent $NO_y^*$ (not enhancement-corrected) as a proxy for particulate $HNO_3$ at flight altitude. (d) Cross-section of $HNO_3$ distribution derived from GLORIA and (e) $HNO_3$ distribution simulated by CLaMS. Flight altitude (bold black line), characteristic waypoints (A, B, C), 340 K and 370 K potential temperature levels (MERRA-2, dashed black lines) and 2 PVU level (MERRA-2, black line). (f) Normalized $HNO_3$ and $O_3$ mixing ratios along selected isentropes. The whole flight was carried out in air masses attributed to the sub-vortex region.

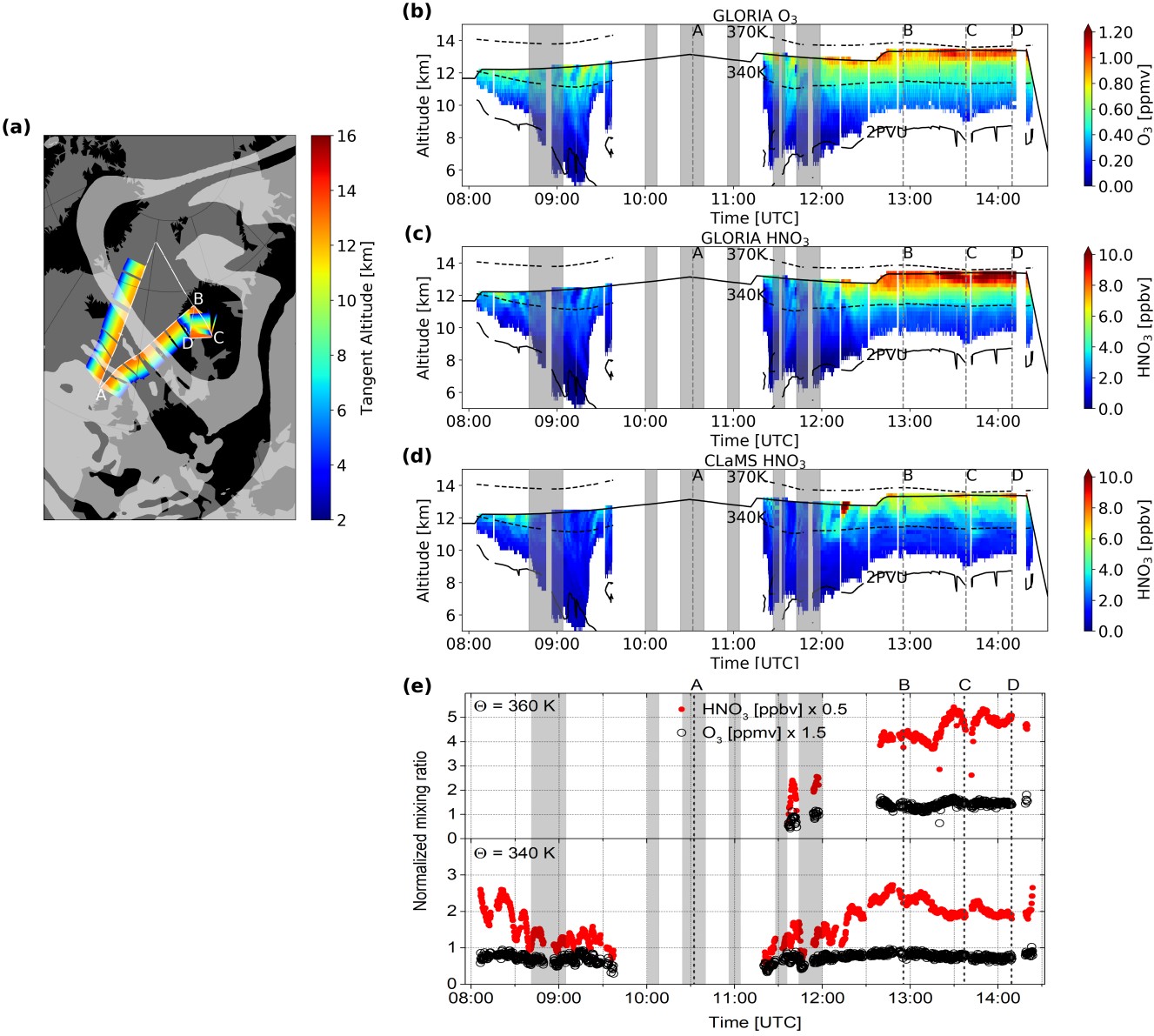

**Figure 3.** (a) Flight path and vortex filtering according to the Nash criterion at 370 K for flight 12 on 31 January 2016. White line: flight track with characteristic waypoints (A, B, C, D); dark grey shading: areas that are associated with the polar vortex. Cross-sections of (b) $O_3$ and (c) $HNO_3$ distribution derived from GLORIA and (d) $HNO_3$ distribution simulated by CLaMS. Flight altitude (bold black line), characteristic waypoints (A, B, C, D), 340 K and 370 K potential temperature levels (MERRA-2, dashed black lines) and 2 PVU level (MERRA-2, black line). Please note the changed colorbar for $HNO_3$ compared to Figs. 2,4. (e) Normalized $HNO_3$ and $O_3$ mixing ratios along selected isentropes. Passages attributed to non-sub-vortex air are shaded in grey.

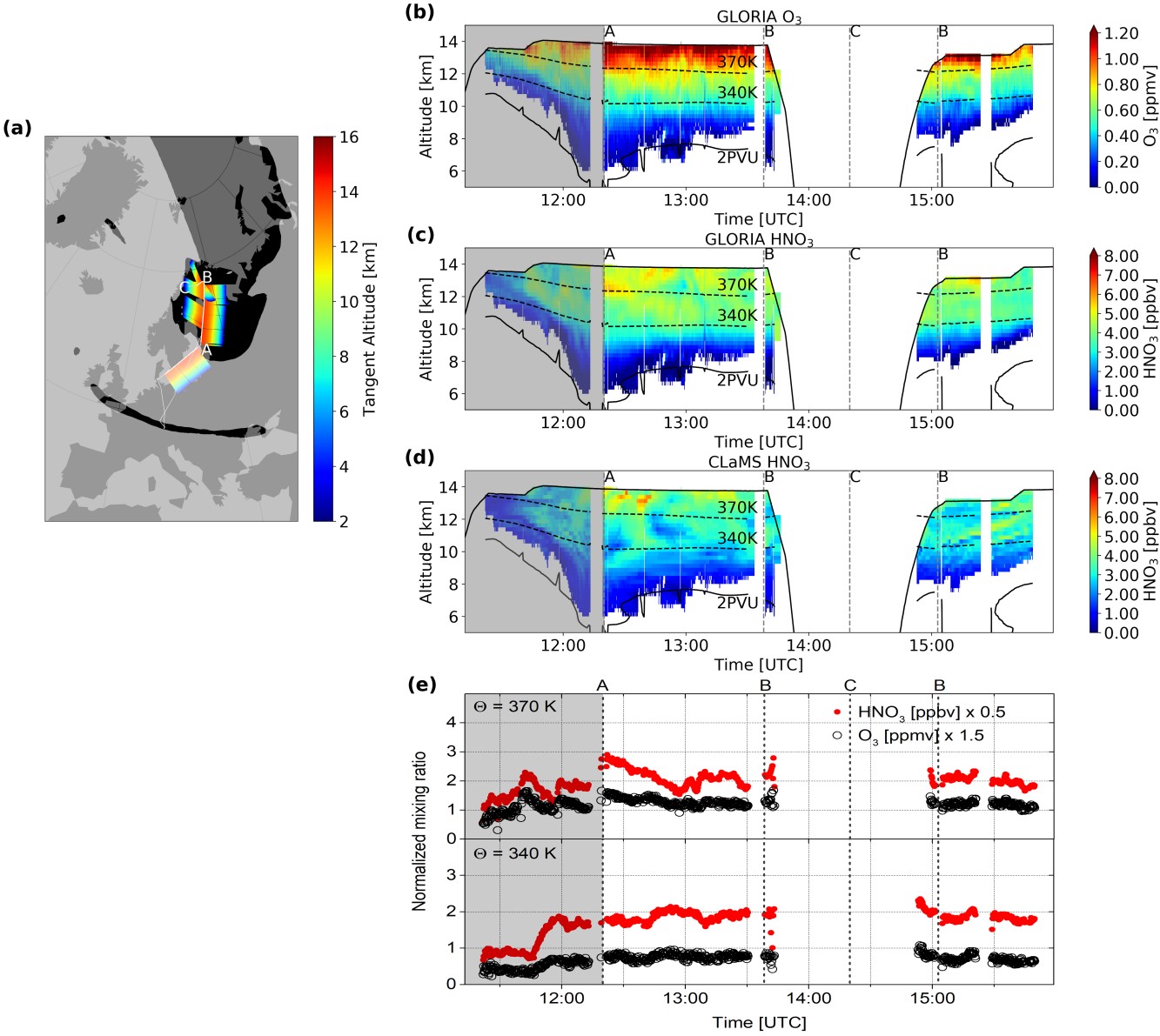

**Figure 4.** (a) Flight path and vortex filtering according to the Nash criterion at 370 K for flight 21 on 18 March 2016. White line: flight track with characteristic waypoints (A, B, C); dark grey shading: areas that are associated with the polar vortex. Cross-sections of (b) $O_3$ and (c) $HNO_3$ distribution derived from GLORIA and (d) $HNO_3$ distribution simulated by CLaMS for flight 21. Flight altitude (bold black line), characteristic waypoints (A, B, C), 340 K and 370 K potential temperature levels (MERRA-2, dashed black lines) and 2 PVU level (MERRA-2, black line). (e) Normalized $HNO_3$ and $O_3$ mixing ratios along selected isentropes. The passage attributed to non-sub-vortex air is shaded in grey.

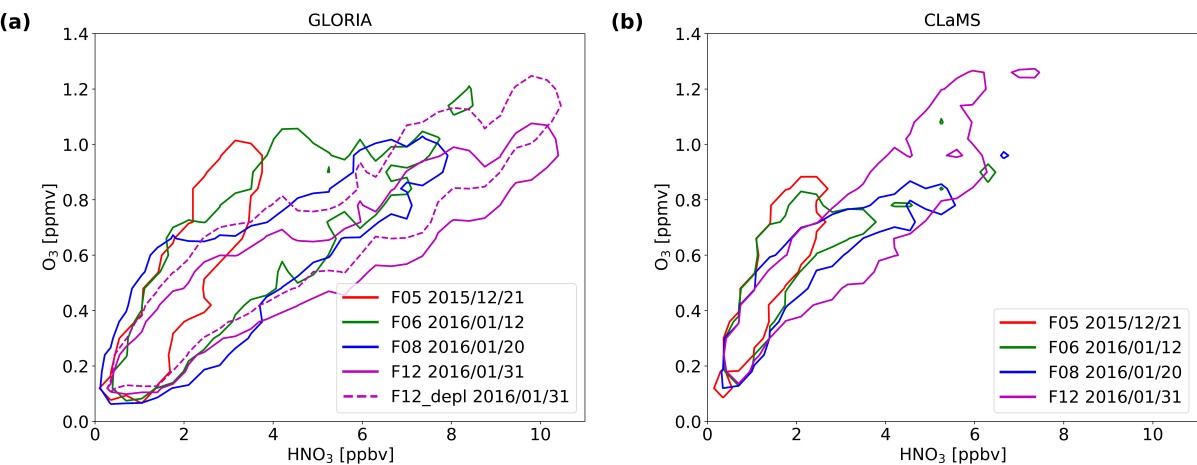

**Figure 5.** Isolines (contours at 0.02) of the normalized frequency distribution of the $O_3$-$HNO_3$-correlation for December 2015 - January 2016 derived from (a) GLORIA measurements and (b) CLaMS simulations. The dashed line for flight 12 includes a compensation of a potential ozone loss of 15 %.

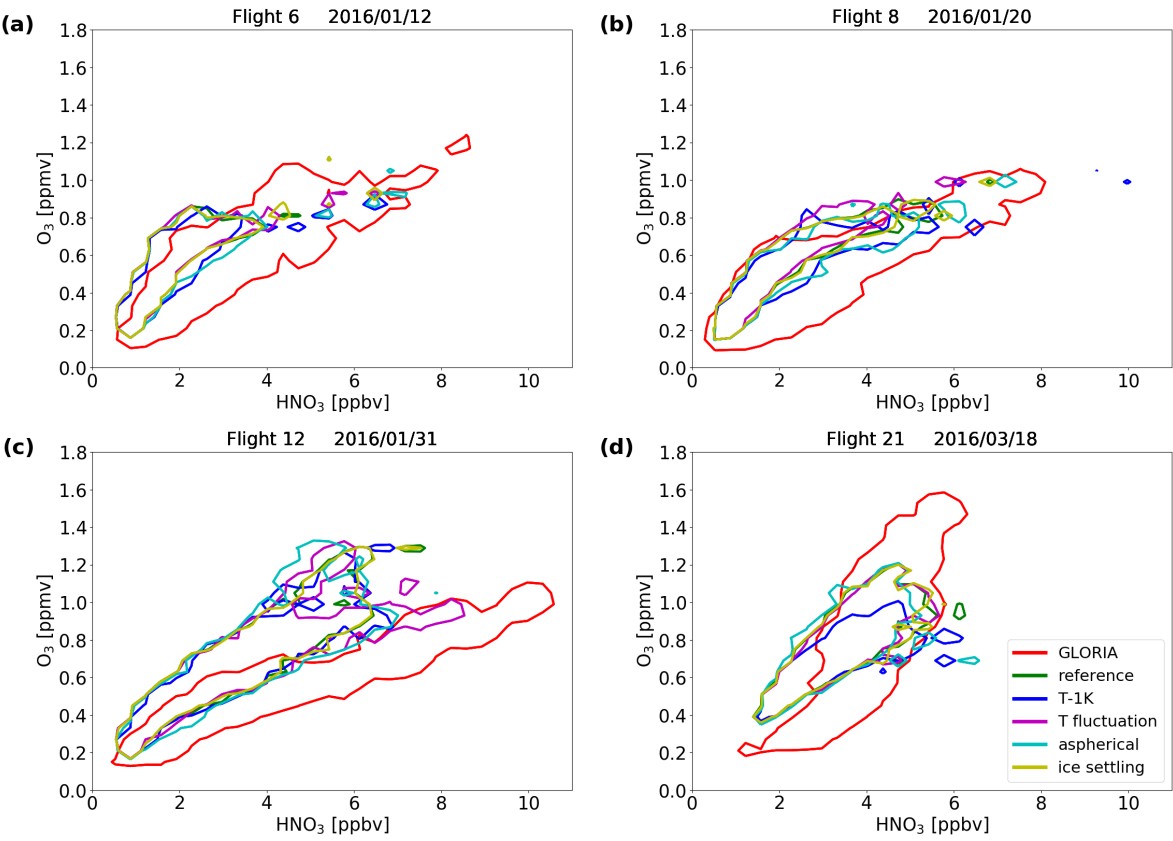

**Figure 6.** Isolines (contours at 0.02) of the normalized frequency distribution of the $O_3$-$HNO_3$-correlation for (a) flight 6 on 12 January 2016, (b) flight 8 on 20 January 2016, (c) flight 12 on 31 January 2016, (d) flight 21 on 18 March 2016 derived from GLORIA measurements (red) and CLaMS simulations.The reference in the panels corresponds to model reference run without perturbations.

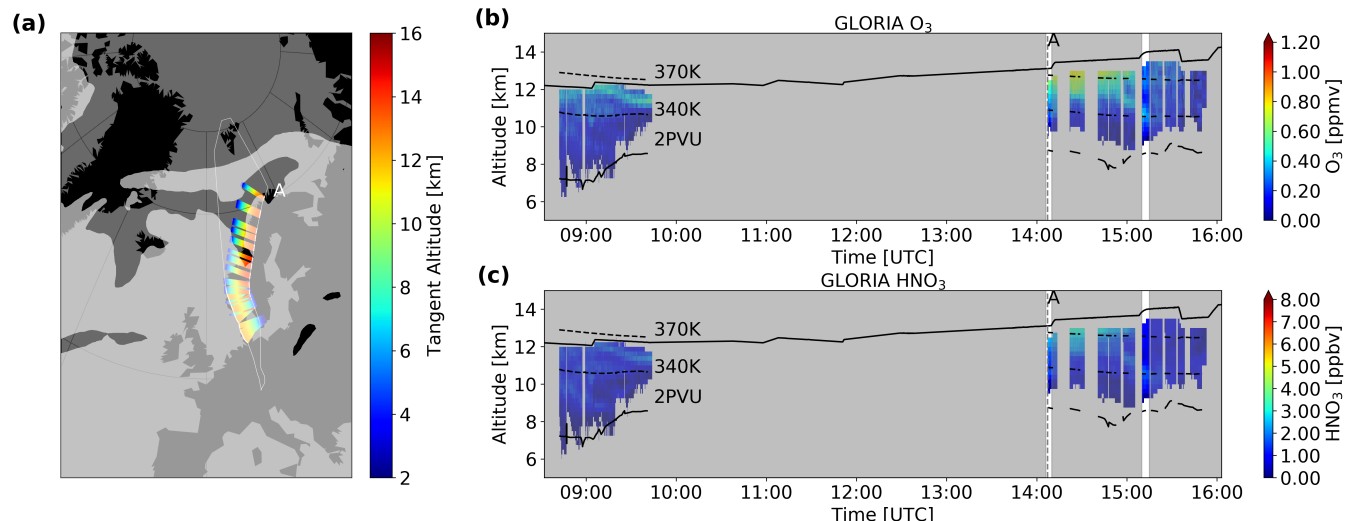

**Figure A1.** (a) Flight path and vortex filtering according to the Nash criterion at 370 K for flight 5 on 21 December 2015. White line: flight track with characteristic waypoint (A); dark grey shading: areas that are associated with the polar vortex. Cross-sections of (b) $O_3$ and (c) $HNO_3$ distribution derived from GLORIA. Flight altitude (bold black line), characteristic waypoint (A), 340 K and 370 K potential temperature levels (MERRA-2, dashed black lines) and 2 PVU level (MERRA-2, black line). Passages attributed to non-sub-vortex air are shaded in grey.

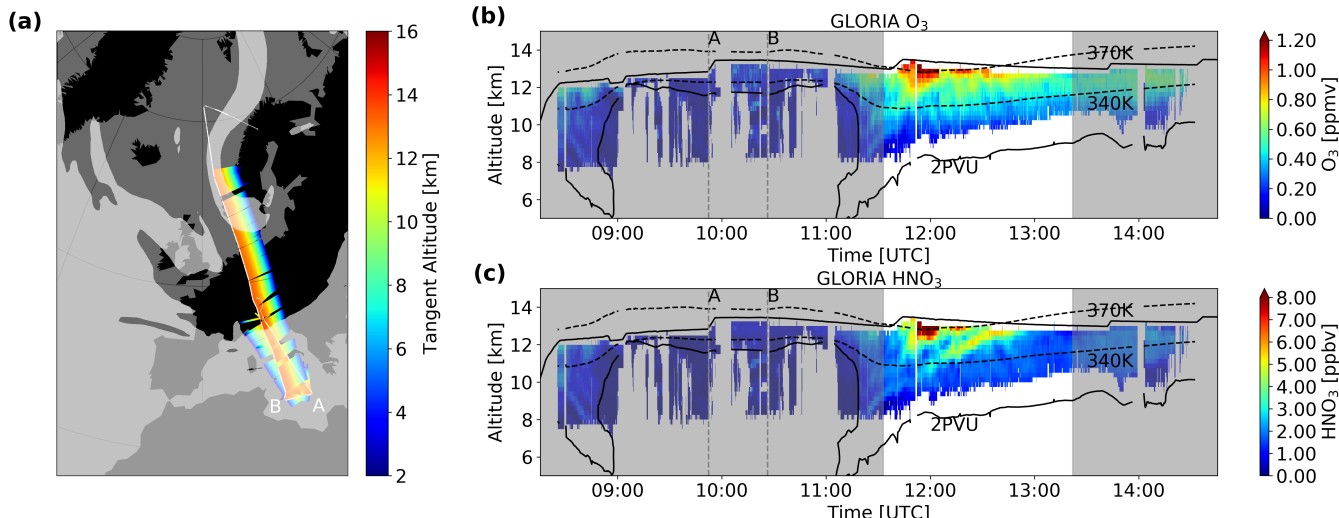

**Figure A2.** (a) Flight path and vortex filtering according to the Nash criterion at 370 K for flight 6 on 12 January 2016. White line: flight track with characteristic waypoint (A,B); dark grey shading: areas that are associated with the polar vortex. Cross-sections of (b) $O_3$ and (c) $HNO_3$ distribution derived from GLORIA. Flight altitude (bold black line), characteristic waypoints (A,B), 340 K and 370 K potential temperature levels (MERRA-2, dashed black lines) and 2 PVU level (MERRA-2, black line). Passages attributed to non-sub-vortex air are shaded in grey.

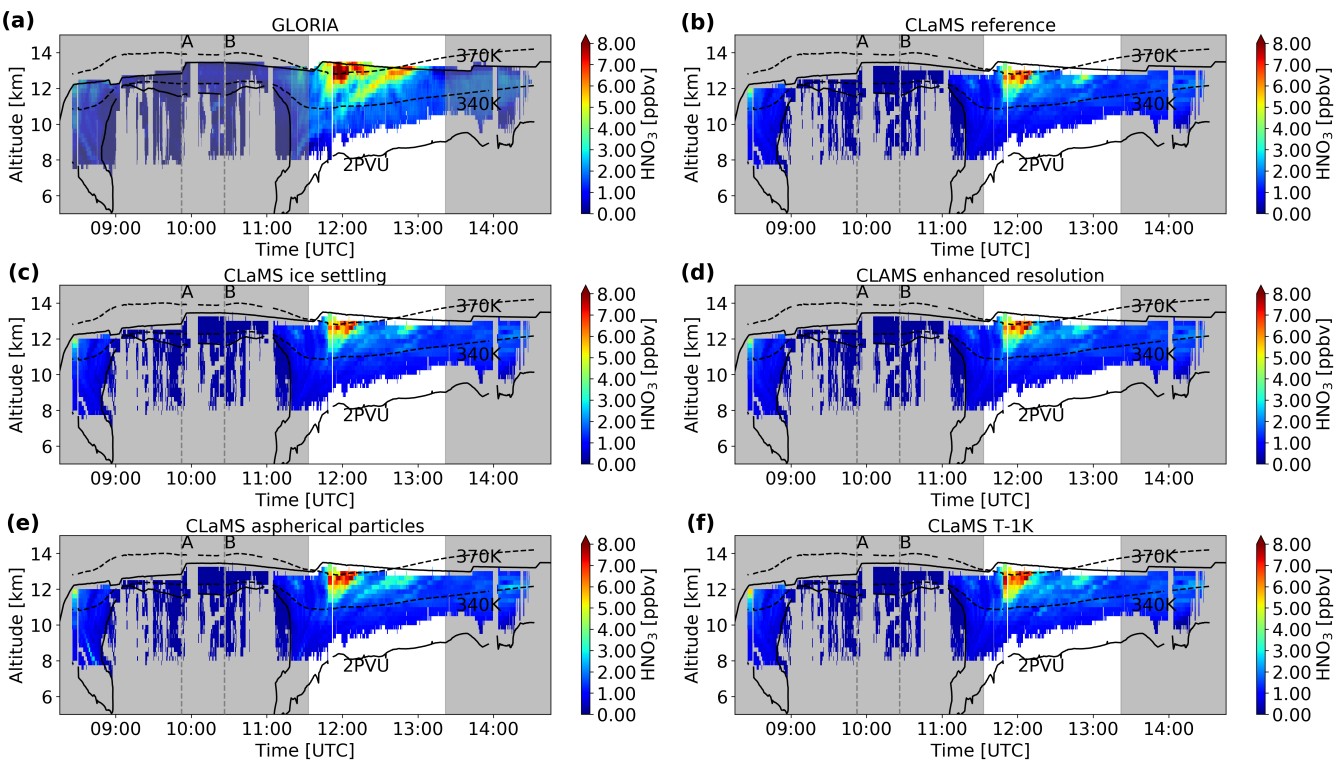

**Figure B1.** Cross-sections of HNO$_3$ volume mixing ratio distribution for flight 6 on 12 January 2016 derived by GLORIA (a) and modelled by the CLaMS reference simulation (b) and sensitivity simulations considering (c) ice formation on NAT particles, (d) temperature fluctuations, (e) growth rates of aspherical particles, (f) 1K global temperature offset. Flight altitude (bold black line), characteristic waypoints (A, B, C), 340 K and 370 K potential temperature levels (MERRA-2, dashed black lines) and 2 PVU level (MERRA-2, black line).

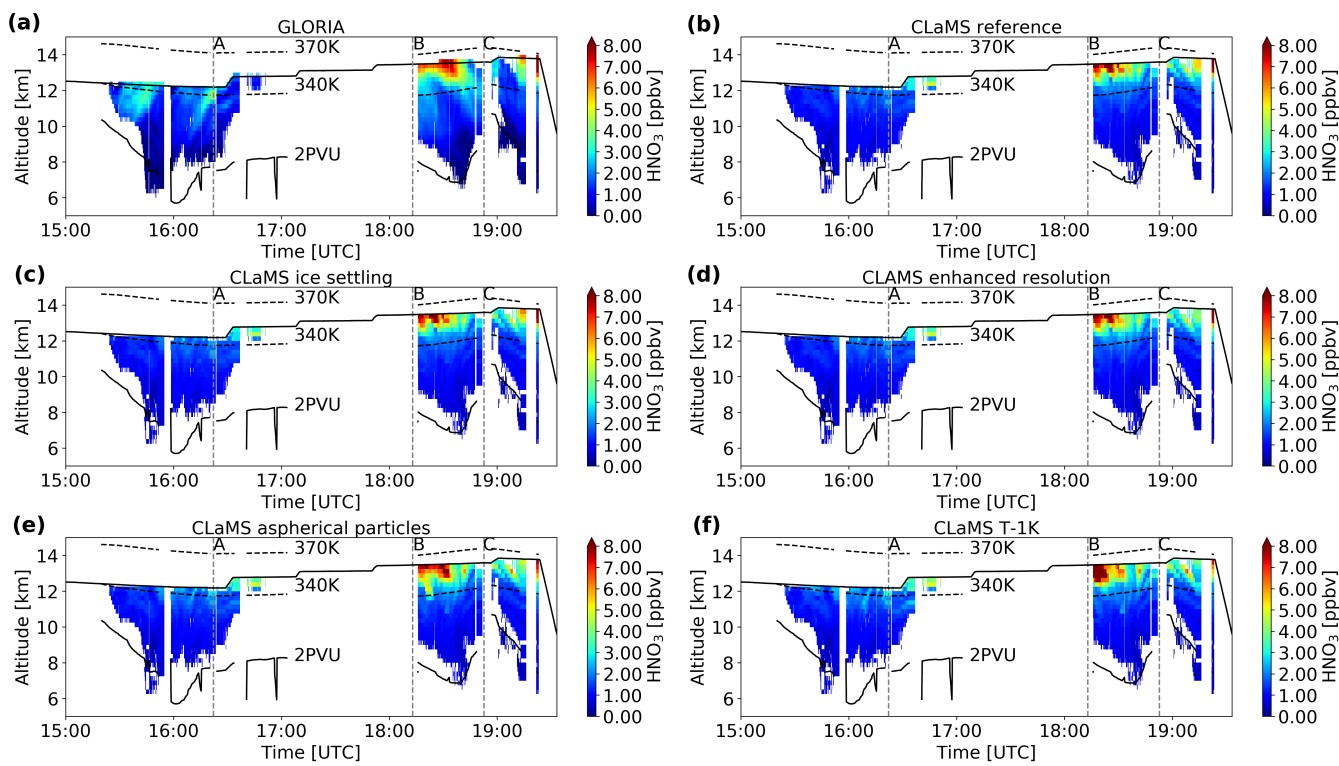

**Figure B2.** Cross-sections of HNO$_3$ volume mixing ratio distribution for flight 8 on 20 January 2016 derived by GLORIA (a) and modelled by the CLaMS reference simulation (b) and sensitivity simulations considering (c) ice formation on NAT particles, (d) temperature fluctuations, (e) growth rates of aspherical particles, (f) 1K global temperature offset. Flight altitude (bold black line), characteristic waypoints (A,B,C), 340 K and 370 K potential temperature levels (MERRA-2, dashed black lines) and 2 PVU level (MERRA-2, black line).

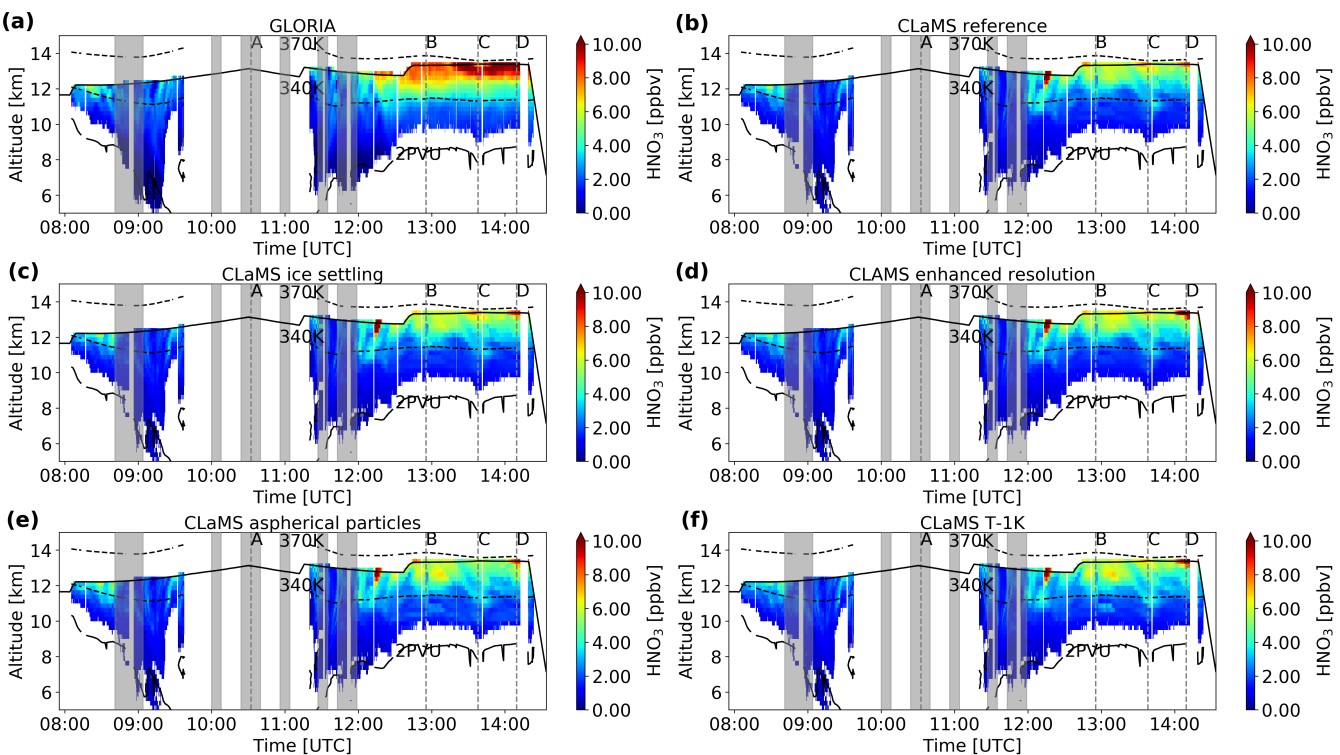

**Figure B3.** Cross-sections of HNO$_3$ volume mixing ratio distribution for flight 12 on 31 January 2016 derived by GLORIA (a) and modelled by the CLaMS reference simulation (b) and sensitivity simulations considering (c) ice formation on NAT particles, (d) temperature fluctuations, (e) growth rates of aspherical particles, (f) 1K global temperature offset. Flight altitude (bold black line), characteristic waypoints (A,B,C,D), 340 K and 370 K potential temperature levels (MERRA-2, dashed black lines) and 2 PVU level (MERRA-2, black line). Please note the changed colorbar compared to Figs. B1,B2,B4.

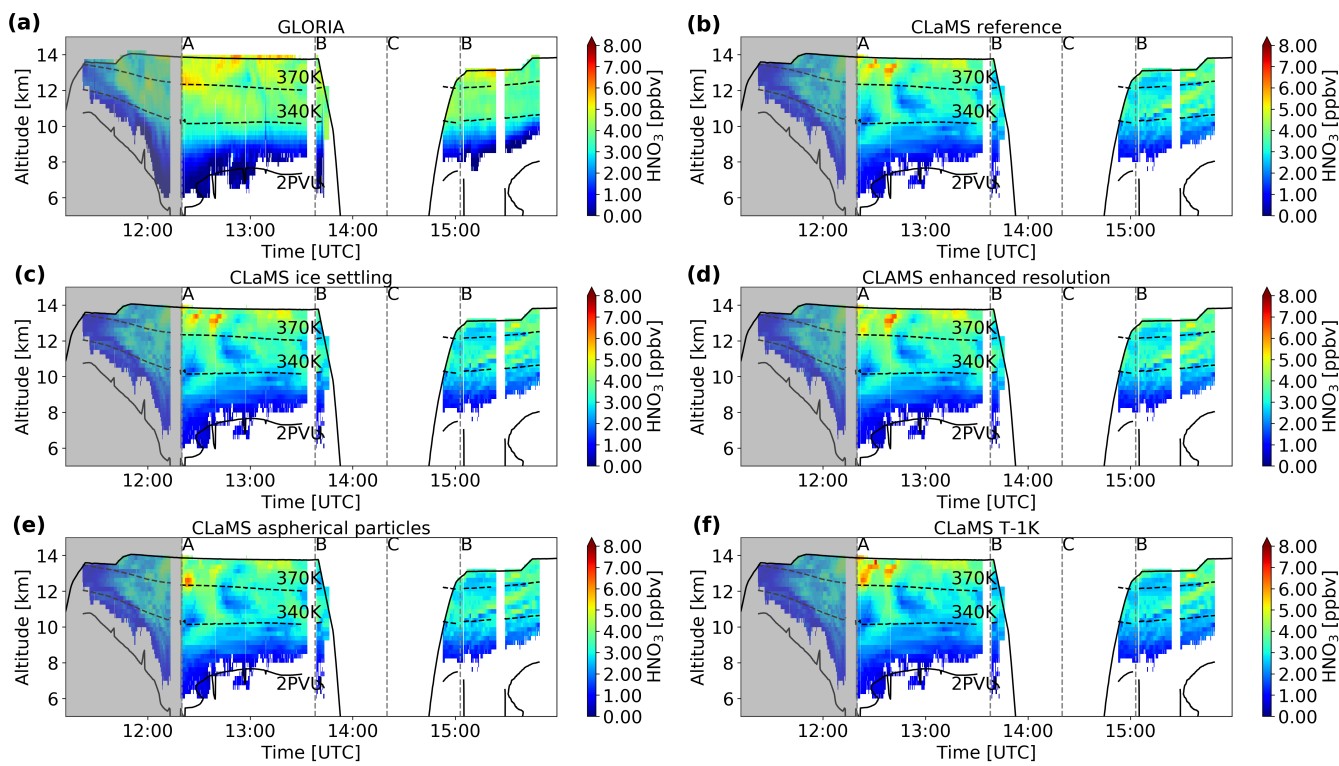

**Figure B4.** Cross-sections of HNO$_3$ volume mixing ratio distribution for flight 21 on 18 March 2016 derived by GLORIA (a) and modelled by the CLaMS reference simulation (b) and sensitivity simulations considering (c) ice formation on NAT particles, (d) temperature fluctuations, (e) growth rates of aspherical particles, (f) 1K global temperature offset. Flight altitude (bold black line), characteristic waypoints (A, B, C), 340 K and 370 K potential temperature levels (MERRA-2, dashed black lines) and 2 PVU level (MERRA-2, black line).