# Peer review of "Nitrification of the lowermost stratosphere during the exceptionally cold Arctic winter 2015/16"

_Atmospheric Chemistry and Physics, 2019_

## Referee Comment (RC1) · Anonymous Referee #1 · 31 Mar 2019

General Comments: The authors have reported the observed HNO3 (and O3) from ∼8 km up to 14 km from GLORIA during the PGS aircraft campaign took place from December 2015 to March 2016. The unique aircraft data will be useful for the atmospheric chemistry community. They have mainly focused on four flights data and also used a chemical transport model CLaMS to investigate the nitrification of the lowermost stratosphere for Arctic winter 2015/16. It is clearly shown that there are still large variabilities of measured HNO3 (and O3) in the LMS along the flight track and CLaMS seems to simulate HNO3 quite well though the model is not perfect to capture some fine structures and also underestimates the observed HNO3. Therefore, the authors have also done four sensitivity experiments to try to understand the discrepancies. Overall, the manuscript is well structured. The data analysis and model results are reasonable.

[Figure]

However, there are some important messages still missing or misleading in the current version. These need to be clarified.

Specific Comments:

1) Selection of aircraft data. It has been mentioned that 18 research flights were carried out between December 2015 and March 2016, but only five flights data are used. Some of other aircraft data may be not suitable for this work, but the authors have not mentioned why they chose these specific 4-5 flights data?

2) ClONO2. I think the comparison of ClONO2 between GLORIA and CLaMS would help since GLORIA has also measured ClONO2 (Johansson et al., 2018) and CLaMS simulates ClONO2.

3) Abstract is not well written and some key points are not supported anywhere (for example the sentence in Lines 10-11, I am also not sure if the conclusion in the Lines 11-12 is a fair statement because other satellite has measured HNO3 in this region). What are missing in CLaMS when the authors conclude the model underestimates ....(Lines 15-16). What is the implication for this work to improve HNO3 simulation in the lowermost stratosphere though some has mentioned in the Introduction?

4) Section 3.2 (Page 5). I am confused with the description. If CLaMS can save daily output at 12:00 UTC, why it can not save the model output along flight track (time, locations etc)? I am not sure why CLaMS needs to re-run forward/backward trajectory for the flight track though I understand CLaMS is based on trajectory calculations..

5) Results explanation. Sometimes it is very hard to follow. For example, Page 7 Lines 9-10. Maybe the coarse vertical resolution is one factor. That will be easier to confirm by increasing the model vertical resolution in the LMS. Pages 9 and 10: What are the key points here? Sorry it is hard to understand the Lines 1-2 in Page 10. Lines 15-18 in Page 12. Not so sure the points of the estimation of lower limit nitrification (though there is an almost linear relationship from the reference in Figure 5).

6) Sensitivity experiments in the Page 13. The descriptions of the model sensitivity experiments are too general. Some of these can only be understandable by the people who are familiar with CLaMS. For example, 'ice settling' simulation, the authors just have one extra criteria to consider in the model (Line 17-18), but we don't know how settling velocity is calculated in the standard CLaMS model . 1.5 times settling velocity for the whole altitude range or something like that needs to add. For the temperature offset, why decrease global temperature by 1K rather than 1.5 or 2 K? Just simple say "NAT formation is T dependent" seems not enough.

7) Discussion and Conclusion. Can you add more why the nitrification for Arctic winter 2015/16 has much more than previous work as you mentioned in the Lines 20-26 in the Page 17?

Technical corrections: 1) Abstract, Page 1 Line 1, change "cold" to "low". 2) Page 1 Line 5, why it is only spatial resolution? Does high temporal resolution matter for this case? 3) Page 1 Line 9. Are you sure about 11 pbbv of HNO3 is observed at 11 km from GLORIA? The only one I can see from Figures 4 and 5 but it occurs above 12 or 13 km (?)

4) Page 3 Line 7-8. What do you mean "mesoscale temperature is not well known"? 5) Page 3 Lines 18-20. This is too general. 6) Page 4 Line 2. "spectra and spectra"? 7) Page 4 Line 25-26. A reference is needed. Is the same reference as Tritscher et al. (2018)? 8) Page 5 Line 20. Add a reference for MERRA2. Why not to use ECMWF ERA interim because you have also done the model simulations based on the meteorological conditions. 9) Page 5 Line 22. Better to use "x" rather than . after "1.2" 10) Page 6 Line 3. Better to add an altitude range after 1.2 ppmv. 11) Page 7 Lines 8-9. Can you make "the enhancement at low altitude" clear? Is it enhancement of HNO3 inside the vortex region compared with outside vortex. Or you mean 2-3 ppbv HNO3 inside the vortex. 12) Page 12. The unit in the text should be consistent with the figure.

---

## Short Comment (SC1) · 31 Mar 2019

This is a thorough and interesting paper on an important topic. I have some brief comments: the phenomenon has been observed in the Antarctic too, consistent with the notion that the conditions in the outer vortex there in 1987 resembled the inner Arctic vortex as regards potential for PSC formation. The effect was indeed observed in https://doi.org/10.1029/GL017i00453 from the DC-8 in the Arctic winter of 1988/89. Its occurrence in the Antarctic is discussed in https://doi.org/10.1002/qj.49712353702.

---

## Referee Comment (RC2) · Anonymous Referee #2 · 29 Apr 2019

This paper presents interesting observations of hno3 from aircraft observations in the Arctic winter of 2016 indicating nitrification of the Arctic polar vortex in the 10-014 km range, the maximum altitude of the observations. While the observations are of interest and the overall analysis convincing, that nitrification of the lower most stratosphere occurred, the paper is poorly written, much too long, and should not be accepted in this form.

Throughout the descriptions of the individual flights, which are used to introduce the observations, the authors claim that the observations show redistribution, enhanced hno3 layers and nitrification. This is before the methods are explained, or the reference profiles discussed. The reader has to guess how these conclusions are made. The description of the figures often resorts to a recitation of numbers in the figures. The

authors specify vortex air in the figures by stating what is not vortex air. Figure caption 2 confuses by not describing the panels in order. Waypoints marked in figures are not used. Reference is made to NOy*, but it is not used further, or defined. The claims of "distinct differences," with some of the CLAMS sensitivity tests, are not well supported by the figures.

The paper could be published, but only after major revision. The analysis should begin by describing that ozone will be used as a tracer for the air sampled and to describe why that works for 2016. Thus the majority of section 5 should appear before any aircraft data are shown. Only two detailed aircraft profiles need to be shown, first the reference profile in December which currently is not shown, and an example of the measurements in January. These two examples are enough to set the stage for the relative normalized frequency distribution (RNFD) discussion, and Figure 6, which is the key figure of the paper. A figure showing an example of RNFDs would also be of interest.

While the authors seem to be keen on showing all of the aircraft data in detail, this really distracts from the main point of the paper. The authors should find another venue to do that and to stick here to the science which can obtained from the data.

Further detailed comments follow by page and line number.

3.6-8 What is meant by stating that "the vertical HNO3 redistribution may be saturated"? How is a vertical redistribution saturated?

5.28-29 and Fig 1 caption. In the text "the identified vortex region (indicated by non-shaded areas in Fig. 1a)" and the figure caption, "light grey shading: areas that are not associated with the polar vortex", are at best confusing and at worst contradictory. The figure caption's version is more consistent with the figure, but then there is a meandering split of the vortex into a western and eastern half. The uniform width of this split seems to indicate more than a filament of non-vortex air.

6.1-2. "Only above the British Isles, southern Scandinavia and north-west of Norway patches of air masses do not fulfil this filter criterion." How is the reader to interpret this statement? Is it an apology that these regions aren't also included as vortex air, thus doubting the Nash criterion? Is "southern Scandinavia" meant to indicate the southern Baltic Sea? The aforementioned region of air nearly splitting the vortex cannot be characterized as "patches of air masses". And with the criterion indicated as vortex a majority of the a/c observations are in this "patch".

6.10-7.2 "In summary, the observed HNO3 structures exhibit a much larger spatial variability than those observed in the ozone distribution, indicating their formation due to redistribution processes." This statement is not acceptable. First the figure does not show a "much larger spatial variability in HNO3 than in ozone. Between 11:30 and 13:00 in the flight data, both gases vary: HNO3 from 3-7 ppbv, a factor of 2, ozone from 0.4 -1.2 ppmv, a factor of 3. Second if there were a difference how does that immediately lead to the conclusion of a redistribution process? There must be some additional explanation to make this leap this early in the paper.

7.11-12 "during in . . ." "Nash criterion, a relatively"

7.13 redundant with 7.11, please don't repeat.

7.15-16 "a number of GLORIA observations where sorted out by cloud-filtering" What does this mean? Sorted out and put where? Do the authors mean remove? I do not understand what sorting out means.

7.20 NOy* has not been explained and there is no reference and it is not used again. Is it important?

7.20-23 What is particulate HNO3? Perhaps these are particles vaporized in an inlet and the hno3 gas measured, or ??? Why are the data not corrected for enhancement efficiency, insufficient information, small correction,. . .? Is it important that they are not corrected? Why do the presence of HNO3 containing PSC particles need to be

confirmed? Confirming compared to what? What other kind of PSC particles could be present besides hno3-containing particles? The first particle maximum doesn't coincided with a GLORIA maximum, but should it, if the hno3 has condensed?

7.30 What is meant by "band-like structures"?

7.33 "distribution, indicating their formation by redistribution of HNO3" The authors again jump to their major conclusion without presenting any reasons.

7.34-35 Here for the first time the authors make an argument for their conclusion, but it is very brief and none of their data has included temperature relative to equilibrium temperatures with respect to NAT. Such information would help the reader understand why in some regions there are particles and in other regions gas phase hno3? In the regions where GLORIA data are shown, should the reader assume these are cloud free?

Figure 2 caption. The panels need to be described in the order in which they appear. A figure caption is so readers can understand what is in the figure. Why confuse them by listing the contents of each panel out of the order in which they appear? Since the interest is in polar vortex air, why do the authors state, here and elsewhere, what is not polar vortex air, rather than what is polar vortex air?

8.1-6 While the hno3 mixing ratios for CLAMS agree with GLORIA, CLAMS does not show anything like the altitude tilted features appearing in the hno3 GLORIA data. What particle information does CLAMS contain? Does that reproduce the in situ particle measurements? Figure 3 Why is waypoint A marked on the map and then not on the panels and not mentioned in the text.

9.8 Why call the hno3 mixing ratios "enhanced" and "strongly enhanced"? This language assumes the authors' pre-determined conclusions prior to the arguments being made. The hno3 mixing ratios are what they are, without this qualifying language.

10.1-2 "Since those structures between waypoints C and E vary significantly from those

observed in the ozone concentrations they most likely originated from nitrification" Is this now the argument to be pursued, using ozone as a tracer? But in fact Figure panels 3b) and 3c) do not support the statement. The structure and the relative magnitudes of ozone and hno3 are quite similar. How do they "vary significantly"?

10.3-4 Now the CLAMS hno3 is "enhanced" even though the maximums and structure of the high hno3 regions do not match the observations. What is the significance of pointing out the very narrow high regions of hno3 in the CLAMS hno3? The last sentence describes well how the model and observations compare.

11.6-9 The enhanced language is used again and the claim of structures indicative of nitrification, as if this point is obvious for almost all of the hno3 values measured by GLORIA greater than some number. Perhaps if the reader were shown "non-enhanced" measurements of hno3 they may agree with the authors about the language, but all we see are hno3 values in the range of 5-10 ppbv pretty much in every flight segment shown.

11.22- It would be helpful to show some of the relative normalized frequency distributions. These could be more interesting for the reader than so many flight profiles, and the pointing out of small features in the flight profiles, which now may be removed as outliers, when the data analysis is finally explained.

12.5- Here finally information on the reference profile which motivated the previous language about enhancements, etc. is offered to the reader. Considering its importance the paper would be better served by showing this nominal reference data for comparison with the more dynamic data later. I am not sure the point of sentences listing the numbers of the maximums. Tables are good for numbers. Text is good for describing general features of the figures such as the progression of the descent of the hno3 over January and how CLAMS doesn't capture this, nor the magnitude of the hno3.

12.14 An ozone loss of 15% significantly reduces the nitrification, so it needs to be clarified whether this was possible. Was such ozone loss occurring in the LMS? CLAMS

should be able to at least estimate this.

12.16 I thought the measurements were filtered from non-vortex air in several ways. Are we now to assume that these correlation plots may be affected by non-vortex air?

13.4-9 Another paragraph pointing out the numbers which the reader can more easily obtain from the figure, or if they are important could be put in a table. Text should be reserved for something more interesting. This whole paragraph and others like it could be mostly removed and the paper would be better for it.

13.10-12 Yes but CLAMS completely misses the continuing descent of the hno3 observed by GLORIA.

13.27-29 "The comparison is based on the RNFDs depicted for the individual flights in Fig. 6." And that is all there is to say about the sensitivity analysis of CLAMS compared to the flight data? Amazing, after the paragraph above describing all the different scenarios to test the sensitivity of CLAMS, no discussion of the results which indicate that the CLAMS results are almost insensitive to these perturbations. Figure 6 the flight dates should be added to the figure panels for reference later.

13.30-16.6 Without much of a difference observed in the summary RNFD plots for the CLAMS sensitivity tests the authors proceed to discuss the flight 6 cross section and its sensitivity simulation in detail, pointing out fine features/differences in Figure 7. But what is the conclusion reached from this detailed discussion? Line 14.8, "However, the overall structure in the RNFD is similar to the reference simulation." Exactly, which is clear from Figure 6a. With this diversion back to detailed discussions of cross sections I gave up on the paper, assuming the same was going to be done for each subsequent flight. Although this is not done, neither is a general discussion of figure 6 offered. Is it really important to go through each model sensitivity difference for each flight when there are so few differences with the reference? A more helpful discussion of the figure would organize it by model sensitivity, and only discuss those sensitivities which make a significant difference with the reference in the direction of the observations. Based
on Figure 6 this criteria would shorten the discussion considerably. Figure 7 and the current sections 6.1-6.4 should be removed.

17.11-19 Claiming distinct differences is an overstatement. The improvement of the temperature fluctuation for 31 Jan. is only evidenced by increased hno3 near 800 ppbv, otherwise it matches the CLAMS reference and all simulations lay significantly higher than the observations. The improvements in the T-1K simulation are generally hardly outside the reference except for 20 Jan. The aspherical particle case provides only a slight difference on 12 Jan. If some estimates of precision were placed on the CLAMS reference, most sensitivity simulations would be hardly outside. The sensitivity simulations simply do not support the claim of distinct differences. Which sensitivity should be chosen to improve the overall agreement with observations over the campaign? There is none.

---

## Author Comment (AC1) · 11 Jun 2019

We thank Adrian Tuck for his valuable comments, that helped us to improve the manuscript. Our answers are given below. The original comment is repeated in **bold**, changes in the manuscript text are printed in *italic*.

**This is a thorough and interesting paper on an important topic. I have some brief comments: the phenomenon has been observed in the Antarctic too, consistent with the notion that the conditions in the outer vortex there in 1987 resembled the inner Arctic vortex as regards potential for PSC formation. The effect was indeed observed in https://doi.org/10.1029/GL017i00453 from the DC-8 in the Arctic winter of 1988/89. Its occurrence in the Antarctic is discussed in**

**https://doi.org/10.1002/qj.49712353702.**

Thank you for pointing us to this interesting aspect. We added citations of both studies to the manuscript and removed the wording "for the first time" from the abstract and conclusion; while GLORIA in fact provides a broad two-dimensional perspective of nitrification of the LMS for the first time, our wording should not be misinterpreted.

---

## Author Comment (AC2) · 11 Jun 2019

**Answer to Referee Comment 1**

June 11, 2019

We thank referee 1 for valuable comments and suggestions. Our answers are given below. The original referee comment is repeated in **bold**, changes in the manuscript text are printed in *italic*.

**General Comments: The authors have reported the observed HNO3 (and O3) from~8km up to 14 km from GLORIA during the PGS aircraft campaign took place from December 2015 to March 2016. The unique aircraft data will be useful for the atmospheric chemistry community. They have mainly focused on four flights data and also used a chemical transport model CLaMS to investigate the nitrification of the lowermost stratosphere for Arctic winter 2015/16. It is clearly shown that there are still large variabilities of measured HNO3 (and O3) in the LMS along the flight track and CLaMS seems to simulate HNO3 quite well though the model is not perfect to capture some fine structures and also underestimates the observed HNO3. Therefore, the authors have also done four sensitivity experiments to try to understand the discrepancies. Overall, the manuscript is well structured. The data analysis and model results are reasonable.**

We thank referee 1 for this positive statement.

**However, there are some important messages still missing or misleading in the current version. These need to be clarified.**

**Specific Comments:**

**1) Selection of aircraft data. It has been mentioned that 18 research flights were carried out between December 2015 and March 2016, but only five flights data are used. Some of other aircraft data may be not suitable for this work, but the authors have not mentioned why they chose these specific 4-5 flights data?**

We added information on the flight selection in section 2:

*The selection of the flight data was guided by the availability of long continuous "chemistry mode" measurements (see Sect. 2.2) in order to show how patterns in the lowermost stratospheric HNO$_3$ distribution change during the winter. We furthermore focus on flights in January, where PSCs extended down to the LMS and where the most notable changes are found in the observed HNO$_3$ distributions. Since we use ozone as a stratospheric tracer to quantify nitrification, flights in January are preferable since only little chemical ozone loss was diagnosed at this time of the winter when compared to February and March (see Johansson et al., 2019). Further GLORIA "chemistry mode" observations can be found in the supplementary information of Johansson et al. (2018) and at the HALO Database (https://halo-db.pa.op.dlr.de/).*

**2) ClONO2. I think the comparison of ClONO2 between GLORIA and CLaMS would help since GLORIA has also measured ClONO2 (Johansson et al., 2019) and CLaMS simulates ClONO2.**

We agree that a comparison of ClONO$_2$ would be of interesting since this species also contributes significantly to NO$_y$. This aspect is part of the study by Johansson et al. (2019). We added the reference to this study in the manuscript (Page 3, Lines 26-28).

**3) Abstract is not well written and some key points are not supported anywhere (for example the sentence in Lines 10-11,**

Here we refer to P7/L29-34 and P16/L23-24 of the original manuscript in ACPD where the statements in lines 10-11 are discussed.

**I am also not sure if the conclusion in the Lines 11-12 is a fair statement because other satellite has measured HNO3 in this region).**

While we agree that other observations (in situ and satellite, see introduction) observed significant nitrification in the LMS region before, we are not aware of any studies reporting such high levels of HNO3 within the LMS due to nitrification as reported here. However, we would like to mention the separate study by Ziereis et al., which addresses this aspect using in situ observations during the same winter. However, we agree that this statement is somewhat strong and modified the abstract as follows:
*Overall, extensive nitrification of the LMS between 5.0 ppbv and 7.0 ppbv at potential temperature levels between 350 and 380 K is estimated. This extent of nitrification has never been observed before in the Arctic lowermost stratosphere.*

**What are missing in CLaMS when the authors conclude the model underestimates....(Lines 15-16).**

The reasons for this deficiencies in the model are yet unclear. It should be noted that the distribution of tracers depends critically on vertical transport and that the vertical velocities are difficult to model. Another sources of uncertainty are the physical parametrisations and their implementation in the model. Therefore, a point-to-point agreement between model and observations is not expected. However, recent studies (Grooß,2018; Tritscher, 2019) showed that the in principle, the vertical HNO$_3$ (and H$_2$O) redistribution is reproduced well.

**What is the implication for this work to improve HNO3 simulation in the lowermost stratosphere though some has mentioned in the Introduction?**

There is no obvious fix. However, we point out a possible strategy assess our model understanding. We stated in the introduction that we test how well different parameterizations within the model reproduce the GLORIA observations. Based on our results we conclude that the sensitivity simulations show differences and lead to a sometimes improved agreement. However, no sensitivity simulation can be identified that generally improves the model. Therefore, we conclude that more extensive modifications of the model parameters addressed by our study are required and/or that important processes are still missing in the model. For clarification, we modified the abstract at P1/L19ff as follows:

*… waves) slightly improve the agreement with the GLORIA observations of individual flights. However, no parameter could be isolated which results in a general improvement for all flights. Therefore, we conclude that a more comprehensive change in the model representations is required. Still, the sensitivity simulations suggest that details of particle microphysics play a significant role for…*

**4) Section 3.2 (Page 5). I am confused with the description. If CLaMS can save daily output at 12:00 UTC, why it can not save the model output along flight track (time, locations etc)? I am not sure why CLaMS needs to re-run forward/backward trajectory for the flight track though I understand CLaMS is based on trajectory calculations..**

Using a standard forward integration, interpolation would always be required. Regardless how the flight paths are constructed, there would be no guarantee for a trajectory ending on the flight path at the right time. Certainly it could be done online with a slightly improved temporal resolution; however, the standard product for a longer term forward integration is stored in snapshots and processed afterwards.

**5) Results explanation. Sometimes it is very hard to follow. For example, Page 7 Lines 9-10. Maybe the coarse vertical resolution is one factor. That will be easier to confirm by increasing the model vertical resolution in the LMS.**

We investigated this effect and could not confirm the coarse vertical resolution as a dominating factor. However, since this flight was removed in context of suggestions by referee 2, this aspect is not mentioned any more in the revised manuscript.

**Pages 9 and 10: What are the key points here? Sorry it is hard to understand the Lines 1-2 in Page 10.**

We revised the discussion of the vertical cross sections to better highlight the key points (see revised manuscript). Further, we removed the statement given in lines 1-2 (see comments to referee 2).

**Lines 15-18 in Page 12. Not so sure the points of the estimation of lower limit nitrification (though there is an almost linear relationship from the reference in Figure 5).**

For clarification, in the revised figure (now Fig. 4) we constructed the same correlation neglecting a potential ozone loss of 15 % (i.e. the ozone mixing rations are scaled accordingly), which now clearly shows potential impacts of ozone loss on the presented tracer-tracer analysis.

**6) Sensitivity experiments in the Page 13. The descriptions of the model sensitivity experiments are too general. Some of these can only be understandable by the people who are familiar with CLaMS. For example, 'ice settling' simulation, the authors just have one extra criteria to consider in the model (Line 17-18), but we don't know how settling velocity is calculated in the standard CLaMS model. 1.5 times settling velocity for the whole altitude range or something like that needs to add.**

We added a reference for the settling velocity and changed the manuscript for clarification:

*Therefore in the 'ice settling' simulation the computed ice settling velocity (computed as described by Tritscher et al., 2019) was increased by a factor of 1.5 at all locations where the saturation ratio of ice, $S_{Ice}$, is larger than 1.2.*

**For the temperature offset, why decrease global temperature by 1K rather than 1.5 or 2 K? Just simple say "NAT formation is T dependent" seems not enough.**

We chose this sensitivity test based on the study by Hoffman et al., 2017 [ACP 17, 10.5194/acp-17-8045-2017, 2017]. This study found that temperatures from ERA Interim have a bias of 0.8K in comparison with those from Concordiasi long-duration balloon measurements in the Antarctic. For MERRA-2 they find a bias of 1K. We assumed that errors in the Arctic are unlikely to be much larger. Therefore, this number seems useful to us to investigate the impact of a potential temperature bias.

**7) Discussion and Conclusion. Can you add more why the nitrification for Arctic winter2015/16 has much more than previous work as you mentioned in the Lines 20-26 in the Page 17?**

We added following sentences to the manuscript:

*During the Arctic winter 2015/16 exceptionally low stratospheric temperatures occurred and the vortex was sufficiently stable to allow formation of PSCs down to lowest stratospheric altitudes. Those conditions were the prerequisites for the strong nitrification observed and presented here.*

**Technical corrections:**

**1) Abstract, Page 1 Line 1, change "cold" to "low".**

We changed the manuscript according to the referee's suggestion.

**2) Page 1Line 5, why it is only spatial resolution? Does high temporal resolution matter for this case?**

We emphasize the aspect of high spatial resolution, since our measurement technique allows particularly for a high spatial resolution. With regard to temporal resolution, the GLORIA data show continuous observations (i.e. 1 profile is measured in ~13 s), but of different air masses along entire flights or flight sections over periods of several hours. Thus we are focusing on overall conditions that are representative for several hours, while fast developments (i.e. at timescales of seconds or minutes) are not in focus of our study and would require a different experimental/flight design (e.g. "self-match flights" or a slowly moving carrier such as a balloon).

**3) Page 1 Line 9. Are you sure about 11 pbbv of HNO3 is observed at 11 km from GLORIA? The only one I can see from Figures 4 and 5 but it occurs above 12 or13 km (?)**

Thank you for pointing this out. We agree and changed it to 12 km.

**4) Page 3 Line 7-8. What do you mean "mesoscale temperature is not well known"?**

We changed the manuscript to: *mesoscale temperature modulations (e.g. by gravity waves) are…*

**5) Page 3 Lines 18-20. This is too general.**

We changed the manuscript to: *We compare the GLORIA data with simulations by the Chemical Lagrangian Model of the Stratosphere  (CLaMS; Grooß et al., 2014, references therein). To test how well different parametrizations within the same model reproduce the GLORIA observations, four sensitivity studies were performed. Those sensitivity simulations investigated the impact of (i) enhanced sedimentation rates in case of ice supersaturation, (ii) a global temperature offset, (iii) modified growth rates and (iv) temperature fluctuations.*

**6) Page 4 Line 2. "spectra and spectra"?**

For better understanding we rephrased to: *… are transformed into spectra. The spectra from horizontal detector rows …*

**7) Page 4 Line 25-26. A reference is needed. Is the same reference as Tritscher et al. (2018)?**

Yes it is, we added the reference to the manuscript.

**8) Page 5 Line 20. Add a reference for MERRA2. Why not to use ECMWF ERA interim because you have also done the model simulations based on the meteorological conditions.**

We added a reference for MERRA -2. Data products from MERRA-2 were available, too. However, since we assume the MERRA2 and ECMWF datasets to be of comparable quality, we used both datasets.

**9) Page 5 Line 22. Better to use "x" rather than . after"1.2"**

We changed the manuscript according to the referee's suggestion.

**10) Page 6 Line 3. Better to add an altitude range after 1.2 ppmv.**

We removed this flight from the manuscript, based on the suggestions of referee 2.

**11) Page 7Lines 8-9.  Can you make "the enhancement at low altitude" clear?  Is it enhancement of HNO3 inside the vortex region compared with outside vortex. Or you mean 2-3 ppbv HNO3 inside the vortex.**

We removed this flight from the manuscript, based on the suggestions of referee 2.

**12) Page 12. The unit in the text should be consistent with the figure**

We changed the units of the figure to ppmv.

---

## Author Comment (AC3) · 11 Jun 2019

**Answer to Referee Comment 2**

June 11, 2019

We thank referee 2 for valuable comments and suggestions. Our answers are given below. The original referee comment is repeated in **bold**, changes in the manuscript text are printed in *italic*.

**This paper presents interesting observations of hno3 from aircraft observations in the Arctic winter of 2016 indicating nitrification of the Arctic polar vortex in the 10-14 km range, the maximum altitude of the observations. While the observations are of interest and the overall analysis convincing, that nitrification of the lower most stratosphere occurred, the paper is poorly written, much too long, and should not be accepted in this form. Throughout the descriptions of the individual flights, which are used to introduce the observations, the authors claim that the observations show redistribution, enhanced hno3 layers and nitrification. This is before the methods are explained, or the reference profiles discussed. The reader has to guess how these conclusions are made. The description of the figures often resorts to a recitation of numbers in the figures. The authors specify vortex air in the figures by stating what is not vortex air. Figure caption 2 confuses by not describing the panels in order. Waypoints marked in figures are not used. Reference is made to NOy\*, but it is not used further, or defined. The claims of "distinct differences," with some of the CLAMS sensitivity tests, are not well supported by the figures. The paper could be published, but only after major revision.**

We thank referee 2 for pointing out that the presented observations are of interest and for the very helpful and constructive criticism which lead to major changes described under the specific items below.

**The analysis should begin by describing that ozone will be used as a tracer for the air sampled and to describe why that works for 2016. Thus the majority of section 5 should appear before any aircraft data are shown. Only two detailed aircraft profiles need to be shown, first the reference profile in December which currently is not shown, and an example of the measurements in January.**

We agree that the structure of our study is not stated clear enough at the beginning and understand now that our intentions are somewhat difficult to extract. As correctly pointed out by referee 2, the central result is the quantification of nitrification of the LMS ("research question 2"). However, our goal is also to show how 2-dimensional vertical $HNO_3$ distributions in the LMS are structured at different stages of the winter and how these patters compare with an established chemical transport model ("research question 1"). For clarification, we modified the manuscript as follows:

- *How are $HNO_3$ distributions structured in the LMS during the course of the cold Arctic winter 2014/16? How do $HNO_3$ distributions, which are affected by nitrification, compare with the stratospheric tracer ozone? How do observed small-scale spatial patters compare with a model (CLaMS)?*
- *Do tracer-tracer correlations constructed from GLORIA $HNO_3$ and $O_3$ indicate nitrification of the LMS? How does nitrification inferred from the GLORIA observations compare with that inferred from CLaMS? Can we identify a critical model parameter which results in a significant overall improvement of the agreement?*

Thus, our manuscript is structured in the following way: In section 2, we introduce the measurement campaign and the data used. In section 3, we introduce the specific characteristics of the data products and the methods and approach of the analysis.

In section 4, we present GLORIA cross-sections of $HNO_3$ and CLaMS simulations to address the first block of research questions. Thereby, we discuss observed patterns in the $HNO_3$ distribution only qualitatively; here, the underlying assumption is that nitrified regions can be identified as patterns differing qualitatively from patterns in the ozone distribution, which experiences no vertical redistribution associated with PSCs.

In section 5, we finally use the previously introduced GLORIA data to derive nitrification of the LMS using tracer-tracer correlations, and test the sensitivity of the model to modifications of different parameters (second block of research questions).

In our opinion, the discussion of individual flights provides valuable insights into the quality of the data and how nitrification patterns are structured in the LMS. Therefore, we think that the results presented in section 4 are the prerequisite for the analysis presented in section 5.

We agree that the method of quantification of nitrification for research question 2 should be provided already in the methods section and revised the manuscript accordingly. The method is now explained in the new section 3.4.

**These two examples are enough to set the stage for the relative normalized frequency distribution (RNFD) discussion, and Figure 6, which is the key figure of the paper. A figure showing an example of RNFDs would also be of interest. While the authors seem to be keen on showing all of the aircraft data in detail, this really distracts from the main point of the paper. The authors should find another venue to do that and to stick here to the science which can obtained from the data.**

We agree that the presentation of a larger number of flights draws away the reader's attention from the key results and removed flight 6 from the revised manuscript. However, we think that the remaining flights are important to address the first block of research questions. Furthermore, we now provide a rationale for the flight selection (see comments to referee 1). In addition, we added a figure (new Figure 1) showing an exemplary RNFD together with the single correlation points.

**Further detailed comments follow by page and line number.**

**3.6-8 What is meant by stating that "the vertical HNO3 redistribution may be saturated"? How is a vertical redistribution saturated?**

We changed the manuscript to:

*At a later time,  in order to nucleate new NAT particles in denitrified air, lower temperatures are needed because of the already decreased $HNO_3$ mixing ratios. This results in a maximum potential denitrification for a given temperature.  .*

**5.28-29 and Fig 1 caption. In the text "the identified vortex region (indicated by non-shaded areas in Fig. 1a)" and the figure caption, "light grey shading: areas that are not associated with the polar vortex", are at best confusing and at worst contradictory.**

We removed this flight as according to the referee's suggestion.

**The figure caption's version is more consistent with the figure, but then there is a meandering split of the vortex into a western and eastern half. The uniform width of this split seems to indicate more than a filament of non-vortex air.**

We removed this flight as according to the referee's suggestion.

**6.1-2 "Only above the British Isles, southern Scandinavia and north-west of Norway patches of air masses do not fulfil this filter criterion." How is the reader to interpret this statement? Is it an apology that these regions aren't also included as vortex air, thus doubting the Nash criterion? Is "southern Scandinavia" meant to indicate the southern Baltic Sea? The aforementioned region of air nearly splitting the vortex cannot be characterized as "patches of air masses". And with the criterion indicated as vortex a majority of the a/c observations are in this "patch".**

We removed this flight as according to the referee's suggestion.

**6.10-7.2 "In summary, the observed HNO3 structures exhibit a much larger spatial variability than those observed in the ozone distribution, indicating their formation due to redistribution processes." This statement is not acceptable. First the figure does not show a "much larger spatial variability in HNO3 than in ozone. Between 11:30 and13:00 in the flight data, both gases vary: HNO3 from 3-7 ppbv, a factor of 2, ozone from 0.4 -1.2 ppmv, a factor of 3. Second if there were a difference how does that immediately lead to the conclusion of a redistribution process? There must be some additional explanation to make this leap this early in the paper.**

We removed this flight as according to the referee's suggestion.

**7.11-12 "during in..." "Nash criterion, a relatively"**

Done.

**7.13 redundant with 7.11, please don't repeat.**

We have deleted this sentence and added PSC occurrence to 7.13

**7.15-16 "a number of GLORIA observations where sorted out by cloud-filtering" What does this mean? Sorted out and put where? Do the authors mean remove? I do not understand what sorting out means.**

We changed **sorted out** to *removed*.

**7.20 NOy* has not been explained and there is no reference and it is not used again. Is it important?**

We changed "$NO_y*$" to "total $NO_y$"

**7.20-23 What is particulate HNO3? Perhaps these are particles vaporized in an inlet and the hno3 gas measured, or ???**

Yes, NOy containing particles were vaporized and measured as gas phase equivalent.

**Why are the data not corrected for enhancement efficiency, insufficient information, small correction,...? Is it important that they are not corrected?**

They are not corrected because of insufficient information on this parameter for this specific flight configuration. Here we use the data only to show qualitatively that $NO_y$-containing PSC particles were present at flight altitude. This information can be clearly inferred from the data, despite the uncertainty of the inferred absolute values of particulate $NO_y$ which rely on assumptions.

**Why do the presence of HNO3 containing PSC particles need to be confirmed? Confirming compared to what? What other kind of PSC particles could be present besides hno3-containing particles?**

The presence of $HNO_3$-containing particles needs to be confirmed to distinguish from potential cirrus cloud particles. Here, we aim at showing (i) that particles (i.e. a PSC) are indeed found at these low altitudes and (ii) that the particles contain NOy and are therefore likely to be linked with designated nitrification patterns seen in the GLORIA observations.

**The first particle maximum doesn't coincided with a GLORIA maximum, but should it, if the hno3 has condensed?**

In a simplified view, maxima in particulate $NO_y$ should coincide with regions of low gas-phase $HNO_3$ (GLORIA), as particulate $NO_y$ that is transported downward during denitrification from higher layers is still in the solid phase. However, this simplified assumption does not necessarily hold depending on the previous de/nitrification history of the observed air masses. Therefore, here our main point is that we find $NO_y$-containing particles (probably transported downward by gravitational settling during the denitrification of higher layers) in the vicinity of highly structured local maxima of $NO_y$ (which differ from patterns of the "dynamical" tracer $O_3$), which suggests that nitrification was going on in the observed air masses.

**7.30 What is meant by "band-like structures"?**

By band –like structures we referred to the connected/coherent structures tilted with altitude. We changed **band-like structures** to *coherent structures tilted with altitude* in the manuscript.

**7.33 "distribution, indicating their formation by redistribution of HNO3" The authors again jump to their major conclusion without presenting any reasons.**

The indication of nitrification is based on the assumption that ozone and $HNO_3$ are affected by the same dynamical processes. We added an explanation at the beginning of the section to support this conclusion (Page 7, Lines 11-16):

*The observed patterns in the $HNO_3$ distributions are compared with the observed patterns in the ozone distribution. Since ozone and $HNO_3$ are effected by the same dynamical processes, the different patterns in the observed distributions are likely caused by processes that effect only one species (i.e. nitrification due to sublimation of $NO_y$-containing particles sedimented from higher altitudes). Therefore, the local $HNO_3$ enhancements seen in comparing adjacent air masses at a given height level and the deviations of their pattern from the pattern seen in the ozone distribution are interpreted qualitatively as a result of nitrification.*

Although we are convinced that strong hints for nitrification can be inferred from qualitative comparing the GLORIA $HNO_3$ and ozone distributions (which provide the foundation for the subsequent quantitative tracer-tracer analysis), we agree that the main conclusion should not be stated already here and therefore changed "indicated" to "*suggest*".

**7.34-35 Here for the first time the authors make an argument for their conclusion, but it is very brief and none of their data has included temperature relative to equilibrium temperatures with respect to NAT. Such information would help the reader understand why in some regions there are particles and in other regions gas phase hno3? In the regions where GLORIA data are shown, should the reader assume these are cloudfree?**

Prior to the retrieval process of trace gases, the GLORIA data are cloud-filtered according to the cloud index method by Spang et al. (2004) since too strongly influenced spectra by clouds would lead to larger errors in derived trace gas abundances. However, this is a qualitative method for cloud filtering, and under conditions of low cloud index values (i.e. a high influence of particulates in the spectra, as observed here), the presence of cloud particles at flight altitude cannot be excluded. While the particulate in situ $NO_y$ observations (i.e. measured total NOy=(gas phase+particles evaporated in the

instrument; forward inlet) minus gas phase NO$_Y$ (gas phase only, backward inlet) are sensitive to cloud particles, with the GLORIA observations only HNO$_3$ in gas-phase is derived; however, the simultaneous presence of particulate NOy in the same air volume cannot be excluded.

The GLORIA temperature data (not shown, see supplement to Johansson et al., 2018) indicate temperatures close to the equilibrium temperature of NAT (and well above the equilibrium temperature of ice) in the vicinity of the HNO$_3$ maxima and where particulate NO$_Y$ is detected. Thus, the GLORIA data support the processes of evaporation and persistence of NAT particles at temperatures close to saturation versus the NAT phase. However, the accuracy of the GLORIA temperature observations is not sufficient to explain precisely the observed pattern of particulate NO$_Y$ observations and HNO$_3$ maxima.

**Figure 2 caption. The panels need to be described in the order in which they appear. A figure caption is so readers can understand what is in the figure. Why confuse them by listing the contents of each panel out of the order in which they appear?**

We changed this according to the referee's suggestion.

**Since the interest is in polar vortex air, why do the authors state, here and elsewhere, what is not polar vortex air, rather than what is polar vortex air?**

We changed this according to the referee's suggestion.

**8.1-6 While the hno3 mixing ratios for CLAMS agree with GLORIA, CLAMS does not show anything like the altitude tilted features appearing in the hno3 GLORIA data. What particle information does CLAMS contain? Does that reproduce the in situ particle measurements?**

The setup of the CLaMS run is focused on the stratospheric composition and the resolution below ~9 km altitude is rather poor. This is one of the reasons, why CLaMS does not show detailed structures below ~8 km altitude. The evaporating HNO$_3$ from the simulated particles is then distributed within a vertical large grid box and therefore hardly visible.

**Figure 3 Why is waypoint A marked on the map and then not on the panels and not mentioned in the text.**

Waypoint A is not shown in the panels, since this part of the flight is not shown there. We added this information in the figure caption.

**9.8 Why call the hno3 mixing ratios "enhanced" and "strongly enhanced"? This language assumes the authors' pre-determined conclusions prior to the arguments being made. The hno3 mixing ratios are what they are, without this qualifying language.**

We removed the distinction between enhanced and strongly enhanced values.

**10.1-2 "Since those structures between waypoints C and E vary significantly from those observed in the ozone concentrations they most likely originated from nitrification" Is this now the argument to be pursued, using ozone as a tracer? But in fact Figure panels 3b) and 3c) do not support the statement. The structure and the relative magnitudes of ozone and hno3 are quite similar. How do they "vary significantly"?**

When looking closely, different patterns can be identified, e.g. the local HNO$_3$ minimum (~3 ppbv lower compared to ambient mixing ratios) at flight altitude directly around waypoint D which is not identified in the ozone distribution. However, we agree that this aspect is difficult to extract from the figure and does not need to be pointed out here. We therefore removed this statement.

**10.3-4 Now the CLAMS hno3 is "enhanced" even though the maximums and structure of the high hno3 regions do not match the observations. What is the significance of pointing out the very narrow high regions of hno3 in the CLAMS hno3? The last sentence describes well how the model and observations compare.**

As described in the previous section, the "enhancement" is based on the comparison with GLORIA observations of $HNO_3$ mixing ratios in adjacent air masses and at the same potential temperature levels (see potential temperature isolines). Therefore, structured patterns in the $HNO_3$ distributions inside the LMS at the same potential temperature level and differing from the ozone distribution are likely to be affected by an additional non-dynamical process, i.e. nitrification by sublimation of $NO_y$-containing particles originating from higher layers.

In the used measurement mode, GLORIA measures a profile about every 13 seconds resulting in more than 250 profiles per hour. Further, the measurements during this flight span several 1000 km. Therefore, regions that appear narrow in the cross sections are still representing areas of several tens to a few 100 km and should not be interpreted as single data outliers.

**11.6-9 The enhanced language is used again and the claim of structures indicative of nitrification, as if this point is obvious for almost all of the hno3 values measured by GLORIA greater than some number. Perhaps if the reader were shown "non-enhanced" measurements of hno3 they may agree with the authors about the language, but all we see are hno3 values in the range of 5-10 ppbv pretty much in every flight segment shown.**

As described in the previous answer, the "enhancement" is based on the comparison with GLORIA observations of $HNO_3$ mixing ratios in adjacent air and at the same potential temperature levels.

**11.22- It would be helpful to show some of the relative normalized frequency distributions. These could be more interesting for the reader than so many flight profiles, and the pointing out of small features in the flight profiles, which now may be removed as outliers, when the data analysis is finally explained.**

We added an example for the relative normalized frequency distribution to the methods section. However, as we stated before, small features in the vertical cross sections stretching over several profiles are still representing several tens to a few hundreds of kilometers and therefore are not due to outliers.

**12.5- Here finally information on the reference profile which motivated the previous language about enhancements, etc. is offered to the reader. Considering its importance the paper would be better served by showing this nominal reference data for comparison with the more dynamic data later.**

We hope that we could clarify above that the qualitative discussion of nitrification described in the previous sections is not based on this reference profile. The local enhancements described in the previous sections are derived qualitatively by considering modulations in $HNO_3$ distributions on wide ranges at (i) constant potential temperature levels inside the vortex region and (ii) with the corresponding ozone fields. The $HNO_3$ and ozone data shown in the cross-section provide the foundation for the quantitative tracer-tracer correlation discussed here. While the slopes of the different overall relative normalized frequency distributions (RNFDs) indicate how the overall nitrification state of the LMS develops during the winter, the widths of the RNFDs show the variability of the nitrification state at the dates of the flights (i.e. simultaneous presence of already nitrified regions and less/non-nitrified regions; these are the "differences " in the patterns seen when comparing the $HNO_3$ and ozone cross sections).

**I am not sure the point of sentences listing the numbers of the maximums. Tables are good for numbers. Text is good for describing general features of the figures such as the progression of the descent of the hno3 over January and how CLAMS doesn't capture this, nor the magnitude of the hno3.the**

We changed this paragraph to a more descriptive text for better readability.

**12.14 An ozone loss of 15% significantly reduces the nitrification, so it needs to be clarified whether this was possible. Was such ozone loss occurring in the LMS? CLAMS should be able to at least estimate this.**

Our assumption of an ozone loss of 15% is based on the study by Johansson et al. (2019), where the ozone loss during the Arctic winter 2015/16 was estimated based on CLaMS calculations.

Additionally, we investigated the ozone depletion in the LMS based on CLaMS (see following Figure1 (not shown in the manuscript)). This shows a good agreement with the 15% that we assumed in our study.

[Figure]

**Figure 1:** Ozone depletion in the LMS estimated by CLaMS.

For clarification of the given numbers, in the revised figure (now Fig. 4) we constructed the same correlation assuming a potential ozone loss of 15 % (see comments to referee 1).

**12.16 I thought the measurements were filtered from non-vortex air in several ways. Are we now to assume that these correlation plots may be affected by non-vortex air?**

The measurements were filtered such that non-vortex air was excluded, as described in section 3.3. As we pointed out there, a robust identification of vortex air is difficult in the altitude regions where the measurements were made. Therefore an influence by non-vortex air cannot be excluded entirely.

**13.4-9 Another paragraph pointing out the numbers which the reader can more easily obtain from the figure, or if they are important could be put in a table. Text should be reserved for something**

**more interesting. This whole paragraph and others like it could be mostly removed and the paper would be better for it.**

We changed this paragraph to a more descriptive text for better readability.

**13.10-12 Yes but CLAMS completely misses the continuing descent of the hno3 observed by GLORIA.**

Here, the point addressed by the referee is not clear to us. "Descent" in the sense of "nitrification by sedimented NAT particles" is clearly identified in both the GLORIA and CLaMS data as the change of the ozone-HNO$_3$ slopes from flight to flight, but the extent is different in the GLORIA data and the simulation (i.e. less nitrification or "slope change" is found in CLaMS data, but the nitrification is not completely missed). On the other hand, "descent" in the sense of "diabatic airmass descent" cannot be read from this plot, as this diabatic airmass descent would affect ozone and HNO3 in the same way and the correlation slope would remain unchanged.

**13.27-29 "The comparison is based on the RNFDs depicted for the individual flights in Fig. 6." And that is all there is to say about the sensitivity analysis of CLAMS compared to the flight data? Amazing, after the paragraph above describing all the different scenarios to test the sensitivity of CLAMS, no discussion of the results which indicate that the CLAMS results are almost insensitive to these perturbations.**

This statement was used to refer to Figure 6 which is the basis of the discussion in the following sections 6.1 – 6.4.

As suggested by the referee we decided to remove sections 6.1 – 6.4 and offer a more general discussion in section 6 to the reader (see revised manuscript).

**Figure 6 the flight dates should be added to the figure panels for reference later.**

We added the flight dates in Figure 6.

**13.30-16.6 Without much of a difference observed in the summary RNFD plots for the CLAMS sensitivity tests the authors proceed to discuss the flight 6 cross section and its sensitivity simulation in detail, pointing out fine features/differences in Figure 7. But what is the conclusion reached from this detailed discussion?**

As stated before, RNFDs and cross sections offer a different perspective on the data. While RNFDs give a more general picture of the data, the cross-sections give additional information on the representation of spatial fine structures. However, we agree that the observed differences are small and therefore decided to move Figure 7 to the appendix.

**Line 14.8, "However, the overall structure in the RNFD is similar to the reference simulation." Exactly, which is clear from Figure 6a. With this diversion back to detailed discussions of cross sections I gave up on the paper, assuming the same was going to be done for each subsequent flight. Although this is not done, neither is a general discussion of figure 6 offered. Is it really important to go through each model sensitivity difference for each flight when there are so few differences with the reference? A more helpful discussion of the figure would organize it by model sensitivity, and only discuss those sensitivities which make a significant difference with the reference in the direction of the observations. Based on Figure 6 this criteria would shorten the discussion considerably. Figure 7 and the current sections 6.1-6.4 should be removed.**

We agree that a discussion based on model sensitivity would be more helpful and therefore changed the manuscript to a more general discussion of Figure 6.

**17.11-19 Claiming distinct differences is an overstatement. The improvement of the temperature fluctuation for 31 Jan. is only evidenced by increased hno3 near 800 ppbv, otherwise it matches the CLAMS reference and all simulations lay significantly higher than the observations. The improvements in the T-1K simulation are generally hardly outside the reference except for 20 Jan. The aspherical particle case provides only a slight difference on 12 Jan. If some estimates of precision were placed on the CLAMS reference, most sensitivity simulations would be hardly outside. The sensitivity simulations simply do not support the claim of distinct differences.**

We agree that the improvements are small for most of the flights. However, we observe changes for individual flights even though no sensitivity simulation shows a distinct improvement for all flights. Therefore we changed **distinct** to *noticeable*.

**Which sensitivity should be chosen to improve the overall agreement with observations over the campaign? There is none.**

We agree with this statement. This issue is not solved by this study. We stated in the introduction that we test how well different parameterizations within the model reproduce the GLORIA observations. Based on our results, we conclude that the sensitivity simulations show differences and lead to a sometimes improved comparison. However, no sensitivity simulation can be identified that generally improves the model. Potentially, important processes are still missing in the model, or stronger changes of parameters tested here need to be implemented.

---

## Author Response (AR2)

**Reply to Review Report by Referee 2 (Report #1)**

September 2, 2019

*Marleen Braun, on behalf of all co-authors*

We thank Referee 2 for his/her thorough review and helpful comments. Our answers are given below. The original Referee comment is repeated in **bold**, our answers are provided in *italic*, and changes applied to the manuscript in *blue/italic*. Please note: page and line numbers in our replies refer to the manuscript without track changes.

**This paper has improved, but still needs work, as detailed in the following. Thus I do not recommend publication in this form, but encourage the authors to further improve this manuscript.**
*We thank Referee 2 for detailed suggestions for improvement and the encouraging statement.*

**While I don't disagree with the overall conclusions of the paper, I still struggle to find support for the authors' claims of nitrification apparent along isentropes in Figures 3, and 4, as detailed further below. While I remain skeptical of their interpretation of the figures, their claims are eminently testable. Just pick several isentropes in the vortex air sampled and plot normalized ozone and hno3 along each isentrope in another time panel at the base of Figures 2, 3, 4. In a sense this is what the RNFDs are showing, but those do not show variations along isentropes, which the authors repeatedly claim show, qualitatively, nitrification.**
*Our initial intention was to discuss nitrification only qualitatively in context of the vertical cross sections and provide the quantitative analysis of nitrification using the RNFDs. But we understand that the identification of nitrification patterns in the vertical cross sections is difficult for the reader. Thus, we appreciate the very helpful suggestion by Referee 2.*
*Following the Referee's suggestion, we included the following panels showing normalized HNO₃ and O₃ along isentropes into Figures 2, 3 and 4. The discussed nitrification patterns now can be clearly identified (i.e. HNO₃ enhancements versus O₃ when compared to air masses outside the sub-vortex region; local modulations within the sub-vortex region and in its close vicinity).*

*Figure 2f:*

[Figure]

*Figure3e:*

[Figure]

*Figure 4e:*

[Figure]

Following further suggestions by the Referee given below, we now use updated waypoints starting at "A" in all panels. Furthermore, passages not classified as "sub-vortex" are shaded in grey in Figures 2, 3, 4, and in the Appendix.

At the end of the captions of Fig. 2-4 we added
Fig 2: "… (f) Normalized HNO₃ and O₃ mixing ratios along selected isentropes. The whole flight was carried out in air masses attributed to the sub-vortex region."
Fig 3: "… (e) Normalized HNO₃ and O₃ mixing ratios along selected isentropes. Passages attributed to non-sub-vortex air are shaded in grey."
Fig 4: "… (e) Normalized HNO₃ and O₃ mixing ratios along selected isentropes. The passage attributed to non-sub-vortex air is shaded in grey."

At P9/L19, we added:
"… HNO₃. To test this hypothesis, we show normalized GLORIA HNO₃ and O₃ data along selected isentropes in Fig. 2f. Normalisation factors are chosen in a way such that the mixing ratios of both gases are close to 1 in air masses which are not affected by nitrification. In unaffected air masses, the normalized mixing ratios of these gases are expected to show the same pattern. In nitrified air masses, locally enhanced HNO₃ mixing ratios and different modulations are expected relative to O₃. In fact, such local maxima in the HNO₃ can be identified at 340 K around 15:45, 16:25, 18:20 and 18:40 UTC and, more pronounced, at 350 K around 18:35 and after 19:10 UTC. The maxima clearly coincide with local maxima seen in the HNO₃ cross sections. Thus, the …"

At P10/L4, we added:
"…nitrification. Again, the analysis of normalized HNO₃ and O₃ along the selected isentropes clearly shows enhanced and more variable HNO₃ mixing ratios relative to O₃ inside air masses attributed to the sub-vortex region (Fig. 3e). During … "

At P11/L12 we added:
"… winter. The analysis of normalized HNO₃ and O₃ along isentropes shows enhanced and slightly more variable HNO₃ mixing ratios relative to O₃ in the sub-vortex region and its vicinity, thus supporting that these patterns are remnants of nitrification (Fig. 4e). … "

**Figures 2-4 could be improved, as detailed further below, by clearly marking in the GLORIA panels the regions considered vortex air, and limiting the flight track altitude displays in the a) panel to those regions in the vortex corresponding to the data shown in the subsequent panels.**

Following the Referee's suggestion (see also reply to comment to Figure 2), we removed the GLORIA tangent points of flight legs not shown in the subsequent

*panels from Figures 2a, 3a, and 4a. To clearly identify air masses attributed to the sub-vortex region in the plots versus time, we shaded flight sections which do not fulfill this criterion in grey in the subsequent time panels.*

**2.5-7 "While …" Break this confusing sentence into two, or otherwise modify it to be clearer and so it doesn't have to be read twice/three times.**
*Following the Referee's suggestion, we rephrased as follows:*
*P2/L5-7: "… Waibel, 1999).* Denitrification *at higher layers (i.e. around 16 to 22 km) attenuates fast deactivation of catalytically active chlorine species into the reservoir species chlorine nitrate ($ClONO_2$).* However, *chlorine deactivation can be fostered at lower layers …"*

**3.3-4 This is a pretty definitive statement on NAT nucleation with no reference. Since this is one of the remaining unknown questions concerning PSCs, the authors should temper this statement. Change "can" to "may", and then qualify it by indicating that evidence is still lacking to confirm this.**
*Following the Referee's suggestion, we rephrased as follows:*
*P3/L3: "… nucleation* may *begin …"*
*P3/L4: "… $T_{NAT}$.* However, clear evidence of the precise nucleation conditions of NAT particles is still lacking. *NAT …"*

**5.13 There is no reference for Tritscher et al., 2018.**
*We corrected as follows: "…* Tritscher et al., 2019*. …"*

**5.22 It is unusual to begin referencing figures 2 and more when figure 1 has not been introduced. Is the reader now to skip ahead to these figures?**
*At P5/L22, we replaced "…distributions in Figs. 2a, 3a and 4a. …" by "distributions* discussed in Sect. 4.1-4.3*. …"*

**7.5 In the reference Eckstein et al. (2018) for the RNFD those authors use isolines of 0.9, 0.4, 0.05. Here the authors use a single isoline of 0.02, so if I understand correctly 98% of the correlation points are included. Would the results change if this rather generous isoline were increased to 0.05 or even 0.2?**
*The RFND method was chosen as an objective and practical method for data reduction, since overlays of scatter plots in Figure 5 instead of isolines would be very difficult to read. Thereby, the threshold value was set to 0.02 to exclude statistical outliers the GLORIA data in the magnitude of random uncertainties. Alternatively, the analysis could be done using the 0.01 isoline or even the envelope of all data points (compare Figure 1) and would return essentially the same results.*
*A lower threshold value is not useful here, since the majority of the data points is often located at altitudes (or corresponding $O_3$ mixing ratios) outside the region of primary interest (see Figure 1). Thus, a higher threshold value would filter out valuable significant data points in the region of interest.*

*In Figure 1, we added 1 % and 4 % isolines for comparison. Furthermore, we added the following explanation at P7/L6: „…Figure 1.* We also show alternative

*isolines including 1 % and 4 % of the maximum density of the histogram to visualize weaker and stronger thresholds for statistical outliers. In all cases, the isolines show a similar pattern in general. However, stronger threshold values limit the vertical range of the analysis and filter out valuable significant data points."*

[Figure]

*Figure 1. Correlation of $HNO_3$ with $O_3$ measured by GLORIA during flight 8 on 20 January 2016. The correlation is shown as single points and as RNFD (solid, dashed, and dash-dotted lines; see legend).*

**7.8 "In order" is used way too much. Here to even start a sentence, and these two words never improve a sentence. Just leave them out here, and elsewhere. The sentences will be clearer.**

*We deleted "in order" at P3/8, P4/L14, P6/L9, and P7/L8.*

**Figure 2. I don't understand the reason to show the entire flight track in Fig 2a) for the day. Why not just show the portion of the flight track which relates to Figs 2b, c, d, e)? Particularly since the flight doubles back on itself, so it is almost impossible to identify the leg corresponding to the data after way point D. This would also eliminate the need for waypoint A. Or one could show the flight track, but not the tangent point altitudes except in the places where the observations are accepted as cloud free. That is the information desired by the reader.**

*See above: In Figures 2a-4a, we removed the tangent points of flight legs which are not analysed in the subsequent panels and start referencing the waypoints used with "A".*

**9.4-5 It would be of interest for the reader to know that the authors are discussing particle enhancement efficiency and which size of particles may be enhanced, rather than that Fig. 2c) is "showing the measurements on 20 January 2016." The reader already knows that since the discussion is about Figure 2, which has already been introduced as presenting the measurements on 20 January 2016.**

*Following the Referee's suggestion, we deleted "In Fig. 2c, we show... 2016... (Ziereis et al., 2004)." and rephrased P9/L4-5 to discuss the in situ observations and enhancement efficiency in more detail:*

*"...The measurements shown in Fig. 2c are based on the sub-isokinetic sampling of particles with a forward looking inlet (e.g. Fahey et al., 1989; Ziereis et al., 2004). Particles larger than a few tenths of a micrometer are sampled with enhanced efficiency and are detected as gas-phase equivalent $NO_y^*$.*

*The efficiency factor depends among others on the ratio between aircraft and sampling velocity, pressure, temperature, and particle size (e.g. Fahey et al., 1989; Feigl et al. 1999). Maximum enhancement factor may be achieved for particle sizes larger than about 10 µm and is on the order of several tens, depending on the actual combination of the above mentioned parameters. Here, only equivalent $NO_y^*$ that was not corrected for enhancement is shown as a qualitative proxy for particulate $HNO_3$. As...”*

*In the caption of Figure 2, we stated more clearly at P8L/4:*

*“... (c) In situ measurements of gas-phase equivalent $NO_y^*$ (not enhancement-corrected) as a proxy for particulate $HNO_3$ at flight altitude. (d) ...”*

**9.7 Please specify where the local HNO3 maximum detected by Gloria is. It is not obvious to this reader that the particles are only associated with this.**

We specified as follows:

P9/L7: “... GLORIA at waypoint A and after waypoint B.”

**9.26 Suggest to add to this sentence the following. “… by CLAMS, and CLAMS completely misses the vertical structure in the HNO3 measurements.”**

*We do not agree that CLaMS completely misses the vertical structure. After waypoint B, the overall vertical structure is at least qualitatively comparable with the GLORIA observations. Therefore, we added the following more moderate statement:*

P9/L26: “... CLaMS, and CLaMS mostly misses the vertical fine structure.”

**9.33-34 Readers would appreciate it if the non-vortex regions were clearly marked on Figs 2b, c, d).** *This is not applicable here, since the whole flight was carried out in air masses attributed to the sub-vortex region. This is now clearly stated in the modified caption of Figure 2 (see above).*

*In Figures 3 and 4, we now marked the non-vortex regions in the panels Fig. 2b-f, 3b-e, and 4b-e.*

**Figure 3 and 10.2-5 "Compared to the ozone values only varying slightly along an isentrope, the HNO3 volume mixing ratios show larger variations at levels of constant potential temperature …" This statement is not supported by Fig. 3c). Once the regions between 8:40-9:20 and 11:30-11:50 are eliminated, the majority of the HNO3 measurements are relatively flat along the 340 K isentrope (12:50-14:00), the only isentrope shown within the data regions. In fact the ozone and hno3 distributions are not that dissimilar. Thus I have to respectfully disagree with the authors' conclusions that the data show local enhancements of hno3. Test this by plotting o3 and hno3 along the time line on an isentrope.**

*See above. We followed the highly appreciated suggestion by the Referee and plotted normalized $HNO_3$ and $O_3$ along isentropes. These plots support local enhancements of $HNO_3$ and different modulations relative to $O_3$ also for this flight.*

**11.4-6 and Figure 4 Again readers would appreciate it if the edge/non-vortex regions were clearly marked on the b, c, d) panels. Given the**

**scale of panel a) it is hardly a useful graphic since the flight tracks are so overlain.**
*We marked the non-vortex regions in the panels Fig. 2b-f, 3b-e, and 4b-e.*

**11.8-12 Aside from the hno3 maximum at 370 K, point A, which is barely in the vortex air, the hno3 distribution appears pretty constant along the isentropes, and certainly not more variable than ozone, although the color scales are so different perhaps that is what is misleading? Still I find the following statements to be less than obvious from figures 4b, c). "Ozone values along the isentropes vary only slightly. The measured HNO3 distribution (Fig. 4c) shows a higher variability along the isentropes with local maxima for altitudes higher than 9 km reaching maximum values of up to 6 ppbv at flight altitude embedded in background values of 2 to 3 ppbv." The reader does not see the authors' claims about this flight further tested until Figure 6d), where the RNFD for this flight is shown.**
*See above. We followed the highly appreciated suggestion by the Referee and plotted normalized $HNO_3$ and $O_3$ along isentropes. These plots support local enhancements of $HNO_3$ and different modulations relative to $O_3$ also for this flight.*
*We furthermore moderated the statement at P11/L8: "… shows a* slightly *higher variability …"*

**And indeed the flight is not showing the slopes of hno3 vs o3 that are apparent from the January flights, which lends further question as to the authors statements above. Is this why the March flight was not included in Figure 5?**
*The flight in March was not included, since this flight was carried out after the vortex break-up (see P11/L10-11 and Manney and Lawrence, 2016) and therefore no "undisturbed" vortex air was present any more. Here, the dynamical situation has changed, the filamentary remnants of vortex air are affected by in-mixing of extra-vortex air masses, and the altered correlation of $HNO_3$ and $O_3$ does not add useful information for the scope Section 5/Figure 5 (i.e. quantification of nitrification in "undisturbed" sub-vortex air in December/January).*
*However, we decided to include this flight to address our first research question, see P3/L18: " How are $HNO_3$ distributions structured in the LMS during the course of the cold Arctic winter 2015/16? … How do observed small-scale patterns compare with a model (CLaMS)?"*
*For clarification, we added the following:*
*P11/L10: "… vortex break-up* (Manney and Lawrence, 2016) *resulted …*
*P11/L11: "… previous flights.* Since flight 21 was carried out after the vortex break-up and the correlation of $HNO_3$ and $O_3$ was altered by in-mixing of extra-vortex air, this flight is included in the model comparisons Sect. 6, but not in the quantification of nitrification of the LMS in Sect. 5. *…"*

**12.10-14 Is flight 5 sampling vortex air or non-vortex air and at what altitudes? In which flights is the non-vortex air included? The authors do not say. What range of latitudes and altitudes is covered in flight 6? Does it include vortex and sub-vortex air, or non-vortex air? Since flights 5 and 6 are going to be the reference and were not discussed in detail earlier, additional information characterizing the air masses**

**sampled is required here. Parenthetical clauses longer than a few words should be avoided. If it is important, use a full sentence. In this case the sentence forced this reader to re-read the sentence in which the clause is embedded, to remember how the sentence began.**

*We agree that more information on flights 5 and 6 would be helpful and that the use of vortex and non-vortex data points should be specified more clearly.*

*For clarification, we rephrased P12/L12-14 as follows:*

*"... Due to a limited number of points associated with vortex air and since sub-vortex and non-vortex data points show a compact correlation for flight 5, non-vortex points are also included here to extent the available data. For all other flights, only data points associated with vortex air are used. Flight 6  ..."*

*We added the following new Appendix A (the previous Appendix was renamed to "B"):*

*"**Appendix A***

*Figure A1a shows the flight track and GLORIA tangent of points of flight 5 on 21 December 2015. The flight accessed air masses associated with the sub-vortex and its vicinity in the region around Scandinavia. Figures A1b and A1c show the associated vertical cross-sections of $O_3$ and $HNO_3$ derived from the GLORIA observations.*
*Figure A2a shows the flight track and GLORIA tangent of points of flight 6 on 12 January 2016. The flight crossed the polar front jet stream above Italy and accessed sub-vortex air masses between northern Italy and Scandinavia. The associated vertical cross sections of $O_3$ and $HNO_3$ derived from the GLORIA observations are shown in Figure A2b and A2c.*

[Figure]

**Figure A1.** *(a) Flight path and vortex filtering according to the Nash criterion at 370 K for flight 5 on 21 December 2015. White line: flight track with characteristic waypoint (A); dark grey shading: areas that are associated with the polar vortex. Cross-sections of (b) $O_3$ and (c) $HNO_3$ distribution derived from GLORIA. Flight altitude (bold black line), characteristic waypoints (A), 340 K and 370 K potential temperature levels (MERRA-2, dashed black lines) and 2 PVU level (MERRA-2, black line). Passages attributed to non-sub-vortex air are shaded in grey.*

[Figure]

*Figure A2.* a) Flight path and vortex filtering according to the Nash criterion at 370 K for flight 6 on 12 January 2016. White line: flight track with characteristic waypoint (A,B); dark grey shading: areas that are associated with the polar vortex. Cross-sections of (b) O₃ and (c) HNO₃ distribution derived from GLORIA. Flight altitude (bold black line), characteristic waypoints (A,B), 340 K and 370 K potential temperature levels (MERRA-2, dashed black lines) and 2 PVU level (MERRA-2, black line). Passages attributed to non-sub-vortex air are shaded in grey."

In the same context, we modified P12/L10:
"... Flight 5 (only limited GLORIA data available, see Appendix A) was ..."

And P12/L13:
" ... Flight 6 (see Appendix A) covered ..."

**Figure 5. Why isn't the 18 March 2016 flight, where the authors just claim to see nitrification of the LMS, shown on this figure?**
*See above: The flight in March was not included, since this flight was carried out after the vortex break-up (see P11/L10-11 and Manney and Lawrence 2016) and therefore no "undisturbed" vortex air was present any more. Here, the dynamical situation has changed, the filamentary remnants of vortex air are affected by in-mixing of extra-vortex air masses, and the altered correlation of HNO₃ and O₃ does not add useful information for the scope Section 5/Figure 5 (i.e. quantification of nitrification in "undisturbed" sub-vortex air in December/January).*

**12.16 Didn't the authors just say the December flight was used as an early winter reference? There is no need to repeat.**
*Following the Referee's suggestion, we deleted the following sentence at P12/16: "This flight was used as early winter reference."*

**14.1-2 To be clear change this to. "The model cross-sections compared to measurements for flights…"**
*Following the Referee's suggestion we rephrased P14/L1-2 as follows: "…Fig. 6. The model cross-sections compared to measurements for flights 6, 8, 12 and 21 …"*

**Figure 6. legend, what is meant by "reference" is this the CLAMS model reference with no perturbations, as shown in Figure 5? If so this should be stated either in the legend, "model reference" or the figure caption.**
*Following the Referee's suggestion, we added in the caption of Fig. 6 at P14/L4:*
*"…simulations. The reference in the panels corresponds to the model reference run without perturbations."*

**15.21 patters?**
*We corrected at P15/L21 to "patterns"*
* * *
**Reply to Review Report by Referee 1 (Report #2)**

September 2, 2019

*Marleen Braun, on behalf of all co-authors*

We thank Referee 1 for his/her through review and helpful comments. Our answers are given below. The original Referee comment is repeated in **bold**, our answers are provided in *italic*, and changes applied to the manuscript in *blue/italic*. Please note: page and line numbers in our replies refer to the manuscript without track changes.

**The authors have now clarified and sufficiently addressed the comments made for the first version. So I would like to recommend it to publish in ACP.**
*We appreciate the positive rating and thank Referee 1 for this encouraging statement.*

**Minor comments:**

**1) Page 1 Line 22. This statement "Therefore we conclude that a more comprehensive change in the model representations is required" is too general and may be not necessary in the abstract. However, you can move this to the "Discussion and Conclusion" section.**
*According to the Referee's suggestion, we moved this statement to P16/L8:*
*"… remain. Therefore, we conclude that a more comprehensive change in the model representations is required. However, the …"*

**2) Page 3 Line 24. It would be good to first define the acronyms (CLaMS for example) in the Introduction.**
*Following the Referee's suggestion, we now define CLaMS already above. We modified P3/17:*
*"Using the GLORIA observations and simulations by the Chemical Lagrangian Model of the Stratosphere (CLaMS; Grooß et al., 2014, references therein), we investigate…"*
*Since the acronym CLaMS is now defined already here, we removed the definition at P3/L29.*
**Then The number of positive ions in line 79 (43) is not consistent with Figure 1 (41?).**

*We could not identify the passages and line numbers and therefore assume a copy-paste error.*

3) Page 4 Section 2.1. I would suggest to add a reference for "chemistry mode".
*We added the corresponding reference at P4/L12:*
*"... "chemistry mode" (Friedl-Vallon et al., 2014) used..."*

4) Page 7 Line 9. "ozone depletion is small in that period". better change "that period" to "January".
*Following the Referee's suggestion we modified P6/L22:*
*"... small in January as the ..."*

**Further Corrections by the Authors**

*P1/L9: typo, "12 km" ->"13 km"*
*P8/caption Figure 2: we deleted 2x "for flight 8 on 20 January 2016", since this information is given already at the beginning of the caption*
*P9/caption Figure 3: we deleted 1x "for flight 12 on 31 January 2016", since this information is given already at the beginning of the caption*
*P11/caption Figure 4: we deleted 1x "for flight 21 on 18 March 2016", since this information is given already at the beginning of the caption*

[revised manuscript text omitted]

---

## Author Response (AR3)

**Reply to Report by Referee 2 (Report #1)**

October 10, 2019

We thank Referee 2 for his/her time and further helpful comments. Our answers are given below. The original Referee comment is repeated in **bold**, our answers are provided in *italic*, and changes applied to the manuscript in *blue/italic*. Please note: page and line numbers in our replies refer to the manuscript without track changes.

**5.27 Do the authors mean "before" waypoint B?**

*Here, we assume that the Referee refers to P8/L27. In this part of the sentence, our intention was to refer to the HNO$_3$ maximum seen by GLORIA (after waypoint B) and not to the particulate HNO$_3$ measured in situ. For clarification, we rephrased as follows:*

*P8/L26-28: "The in situ data clearly confirm the presence of HNO$_3$-containing PSC particles at flight altitude and mainly between waypoints A and B, in close vicinity to gas-phase HNO$_3$ maxima detected by GLORIA."*

**8._5 I don't understand what happened to the line numbers in the manuscript. They start over after line 30, but at some random point after line 30. I use _ to indicate I am using the second line numbering system on the page. This is the second manuscript I have seen this on in the last few days.**

*We apologize for this compilation error and corrected this in the updated manuscript.*

**8._8-9 A bit more explanation is required for the normalization. With this method it seems like the normalized mixing ratios of both gases should be one at some point in sub vortex air, and this is the case in Figs. 2 and 3, but not in 4. In Fig. 4 it appears the normalization was done in non-sub-vortex air. Which doesn't seem quite right. Have the authors considered using a climatology of sub-vortex air stratified by season? Could be a lot of work. In any case a bit more explanation would be appreciated.**

*Following the Referee's suggestion we added the following explanation:*

*P8/L_9ff: "… nitrification. In particular, HNO$_3$ volume mixing ratios (in ppbv) were multiplied by a constant factor of 0.5, and O$_3$ volume mixing ratios (in ppmv) by a constant factor of 1.5 to result in the shown unitless normalized mixing ratios. The same factors were applied for the subsequent flights. In unaffected …"*

*P9/L30: (Fig. 3e, for normalization see Sect. 4.1)*

*P10/L18: (Fig. 4e, for normalization see Sect. 4.1)*

**Figures 2f), 3e), 4e) Very nice additions. I have two suggestions: 1 – make the symbol size the same, 2 – include similar CLAMS normalized mixing ratios.**

*1- We changed the symbol size according to the Referee's suggestion. 2- We appreciate the suggestion by the Referee, but we think that in this case the panels would become relatively busy and not much information would be added beyond the comparisons of the GLORIA/CLaMS HNO$_3$ cross-sections and the subsequent quantitative comparisons using the tracer-tracer correlations. Therefore, we would prefer to keep the plots unchanged in this case.*

**9.26 Are these regions indicated by the gray shading? Is so include mention of this.**

*Following the Referee's suggestion we modified P9/L26: "… non vortex air (indicated by grey shading in Fig. 3b-e), ozone …"*

**9.30 Suggest … maximum values of HNO3 well …**

*Following the Referee's suggestion we modified P9/L31: "… maximum values of HNO$_3$ well…"*

**10.28 … extend …**

*We changed this according to the Referee's suggestion.*

[revised manuscript text omitted]